# Emergence of power law distributions in protein-protein interaction networks through study bias

David B Blumenthal[1]*[†], Marta Lucchetta[2†‡], Linda Kleist[3§], Sándor P Fekete[3,4], Markus List[5,6]*, Martin H Schaefer[2]*

[1]Biomedical Network Science Lab, Department Artificial Intelligence in Biomedical Engineering, Friedrich-Alexander-Universität Erlangen-Nürnberg, Erlangen, Germany; [2]Department of Experimental Oncology, IEO European Institute of Oncology IRCCS, Milan, Italy; [3]Department of Computer Science, TU Braunschweig, Braunschweig, Germany; [4]Braunschweig Integrated Centre of Systems Biology (BRICS), Braunschweig, Germany; [5]Data Science in Systems Biology, TUM School of Life Sciences, Technical University of Munich, Freising, Germany; [6]Munich Data Science Institute (MDSI), Technical University of Munich, Garching, Germany

**\*For correspondence:**
david.b.blumenthal@fau.de (DBB);
markus.list@tum.de (ML);
martin.schaefer@ieo.it (MHS)

[†]These authors contributed equally to this work

**Present address:** [‡]Integrated Biology of Rare Tumors, Department of Experimental Oncology, Fondazione IRCCS Istituto Nazionale dei Tumori, Milan, Italy; [§]Institute of Computer Science, University of Potsdam, Potsdam, Germany

**Competing interest:** The authors declare that no competing interests exist.

**Abstract** Degree distributions in protein-protein interaction (PPI) networks are believed to follow a power law (PL). However, technical and study biases affect the experimental procedures for detecting PPIs. For instance, cancer-associated proteins have received disproportional attention. Moreover, bait proteins in large-scale experiments tend to have many false-positive interaction partners. Studying the degree distributions of thousands of PPI networks of controlled provenance, we address the question if PL distributions in observed PPI networks could be explained by these biases alone. Our findings are supported by mathematical models and extensive simulations, and indicate that study bias and technical bias suffice to produce the observed PL distribution. It is, hence, problematic to derive hypotheses about the topology of the true biological interactome from the PL distributions in observed PPI networks. Our study casts doubt on the use of the PL property of biological networks as a modeling assumption or quality criterion in network biology.

## Editor's evaluation

This manuscript makes an important contribution to the understanding of protein-protein interaction (PPI) networks by challenging the widely held assumption that their degree distributions uniformly follow a power law. The authors present convincing evidence that biases in study design, such as data aggregation and selective research focus, may contribute to the appearance of power-law-like distributions. While the power law assumption has already been questioned in network biology, the methodological rigor and correction procedures introduced here help to advance our understanding of PPI network structure.

## Introduction

*Barabasi and Albert, 1999* proposed in the late 1990s that naturally occurring networks have a commonality: The distribution of their node degrees $k$ (i. e. the number of interactions each node is participating in) tends to follow a PL distribution $P(k) \propto k^{-\alpha}$. For $2 < \alpha < 3$, this distribution is scale-free, as its variance diverges with increasing network size. An important consequence of this assumed long-tail distribution of the node degrees is that it explains the existence of hub nodes with

many connections (which are unlikely to occur under other statistical models), contrasting a large number of lowly connected nodes. Another feature of PL-distributed networks is their small world property, where a small network diameter leads to relatively high resilience against random perturbations (*Cohen et al., 2000*). This commonality in the topology of real-world networks is considered a universal law, as it seems to describe common features of such diverse networks such as food webs, metabolic networks, the internet, and (PPI) networks (*Jeong et al., 2001*; *Barabási and Oltvai, 2004*; *Yook et al., 2004*).

With respect to PPI networks, the PL property is typically explained with biological considerations: Protein families that are involved in general biological processes such as protein folding, gene regulation, or post-translational modifications are very promiscuous and bind to a large number of other proteins, whereas the majority of proteins show few interactions (*Nobeli et al., 2009*). Moreover, it is crucial for the emergence of the PL property that, in the evolution of networks, 'new vertices attach preferentially to sites that are already well connected' (*Barabasi and Albert, 1999*). It has been suggested that, in the evolution of PPI networks, such preferential attachment can be explained via gene duplication and subsequent mutation (*Pastor-Satorras et al., 2003*).

Today, the assumption that PPI networks show a PL distribution has been codified in textbooks (*Barabási and Pósfai, 2016*) and training material (*Millán, 2016*). This has had an important implications on the network biology field: Some studies use PL fittings as quality criteria for their measured networks *Stelzl et al., 2005*; others use topological protein properties ex- or implicitly for predicting disease genes (*Xu and Li, 2006*; *Janyasupab et al., 2021*). Further examples are the co-expression module inference tools WGCNA (*Zhang and Horvath, 2005*; *Langfelder and Horvath, 2008*) and CEMiTool (*Russo et al., 2018*). In these tools, the assumption that biological networks are PL-distributed directly informs the automated choice of hyper-parameters used to prune or transform the co-expression matrices, i.e., the hyper-parameters are chosen such that the resulting degree distributions yield good PL fits. Since WGCNA is extremely widely used (more than 17,000 citations according to Google Scholar as of February 2024), the PL assumption has hence potentially shaped the results reported in thousands of studies.

Even though it was reported that PL properties of networks across disciplines often lack statistical support or mechanistic backing (*Stumpf and Porter, 2012*), the assumption that PPI networks are PL-distributed has become mainstream in the network biology field. With respect to PPI networks, critical voices have been raised since the 2000s. Broadly, these studies can be categorized in two groups. Firstly, various studies exist that challenge the correctness of the claim that empirical PPI networks are scale-free or PL-distributed (*Pržulj et al., 2004*; *Tanaka et al., 2005*; *Khanin and Wit, 2006*; *Lima-Mendez and van Helden, 2009*; *Przulj et al., 2010*; *Broido and Clauset, 2019*): In some of these studies, goodness-of-fit tests are used to show that, in some empirical PPI networks, PL distributions actually do not provide a good fit of the empirical degree distributions (*Tanaka et al., 2005*; *Khanin and Wit, 2006*; *Lima-Mendez and van Helden, 2009*; *Broido and Clauset, 2019*). In others, networks are simulated using random network generation models that do and do not yield PL-distributed networks, and it is then argued that the simulated non-PL networks are often more similar to empirical PPI networks than the simulated PL networks (*Pržulj et al., 2004*; *Przulj et al., 2010*).

Second, there are studies which concede (at least for the sake of the argument) that empirical PPI networks are PL-distributed, but challenge that this is sufficient evidence to conclude that the same holds for the ground truth interactome (*Stumpf et al., 2005*; *Han et al., 2005*; *Deeds et al., 2006*): In some of these studies, it is argued that (dis-)appearance of PLs in empirical PPI networks may be artifacts of sampling from the full interactome (*Stumpf et al., 2005*; *Han et al., 2005*). In another study, a physical network generation model is presented, which allows us to explain the emergence of PLs in empirical PPI networks as an artifact of technical biases in yeast-2-hybrid (Y2H) screens (*Deeds et al., 2006*).

In this work, we aim to rekindle interest in a critical assessment of the assumption that PPI networks are PL-distributed and posit biased research interest in proteins as another possible non-biological explanations (*Figure 1A*). Based on data from more than 40,000 affinity purification-mass spectrometry (AP-MS) and Y2H studies, we argue that the emergence of PL distributions in empirical PPI networks can be explained by a combination of the following three factors:

- Study bias in the selection of the tested proteins: PPIs are typically detected using a Y2H screen studies, where individually selected protein pairs or libraries can be tested as bait and prey, or

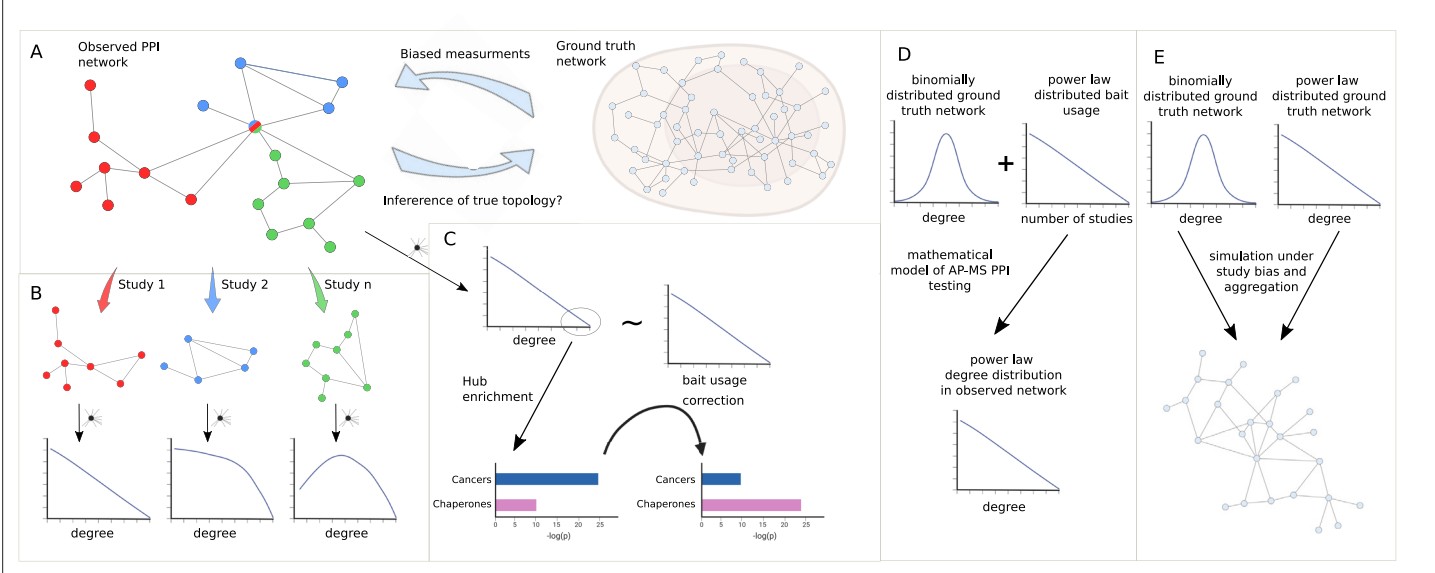

**Figure 1.** Study overview. (**A**) We seek to answer the question of how much we can learn about the topology of ground truth networks from the topology of observed and aggregated protein-protein interaction (PPI) networks and how much-biased measurements might impact the observed power law (PL) degree distribution. (**B**) To answer this question we decompose aggregated, observed networks into single study-networks and investigate their individual degree distributions. We then ask how much the aggregation process of those single studies into larger networks could explain the PL property of the observed network. (**C**) We aim to identify true hub proteins by applying different types of normalization strategies, which reveals that disease-associated functions disappear that are likely associated with hub proteins because of their inflated testing frequency due to the study bias. (**D**) Starting from the empirical observation that bait usage is PL-distributed, we mathematically show that, in such a scenario, a PL-distributed observed PPI network can emerge even if the ground truth is binomially distributed. (**E**) Finally, we simulated the measurement of observed aggregated PPI networks under study bias from ground truth networks with either PL or binomial degree distribution.

via AP-MS, where one or several bait proteins are tested against a large number of preys. In particular AP-MS experiments are sensitive to study bias, where already overstudied proteins such as oncogenes or tumor suppressors are tested more frequently than others (***Schaefer et al., 2015***).

- False positives in the experimental techniques used for measuring PPIs: We know that biases of the experimental procedures used to infer networks can affect the resulting topology (***Peel et al., 2022***). This is particularly relevant for PPI networks, which are based on techniques with an estimated false positive rate of up to 80% (***Berggård et al., 2007***).
- Aggregation of the results of single experiments: Today, researchers in the network biology field mainly rely on aggregated PPI networks obtained from databases such as HIPPIE (***Alanis-Lobato et al., 2017***), BioGRID (***Oughtred et al., 2021***), IID (***Kotlyar et al., 2022***), or STRING (***Szklarczyk et al., 2021***). We show that, in combination with study bias and a non-zero false positive rate, such aggregation can lead to PL distributions in empirical PPI networks even if the measured ground truth interactome has a radically different topology.

To do so, we show that only a subset of networks exhibit a node degree distribution following a PL. We then systematically test if the PL property arises simply by aggregating studies (***Figure 1B***), as is common practice in PPI databases. Next, we test if the node degree distribution still follows a PL if we account for the bias introduced by bait proteins. Furthermore, we test to which extent accounting for such biases changes the functional enrichment of highly promiscuous hub proteins, where we expect that heavily studied disease-related proteins show reduced enrichment whereas functions carried out by proteins known to be promiscuous should show increased enrichment (***Figure 1C***). We then show mathematically that, given PL-distributed bait usage, PL-distributed PPI networks can emerge through aggregated AP-MS testing even if node degrees are binomially distributed in the unknown ground truth interactome (***Figure 1D***). Finally, we simulate the measurement process of observed PPI networks under study bias for different false negative and false positive rates, given hypothetical PL-distributed and binomially distributed ground truth interactomes (***Figure 1E***). Using $K$-nearest neighbors ($K$-NN) classification in the space of degree distributions of the simulated PPI networks, we

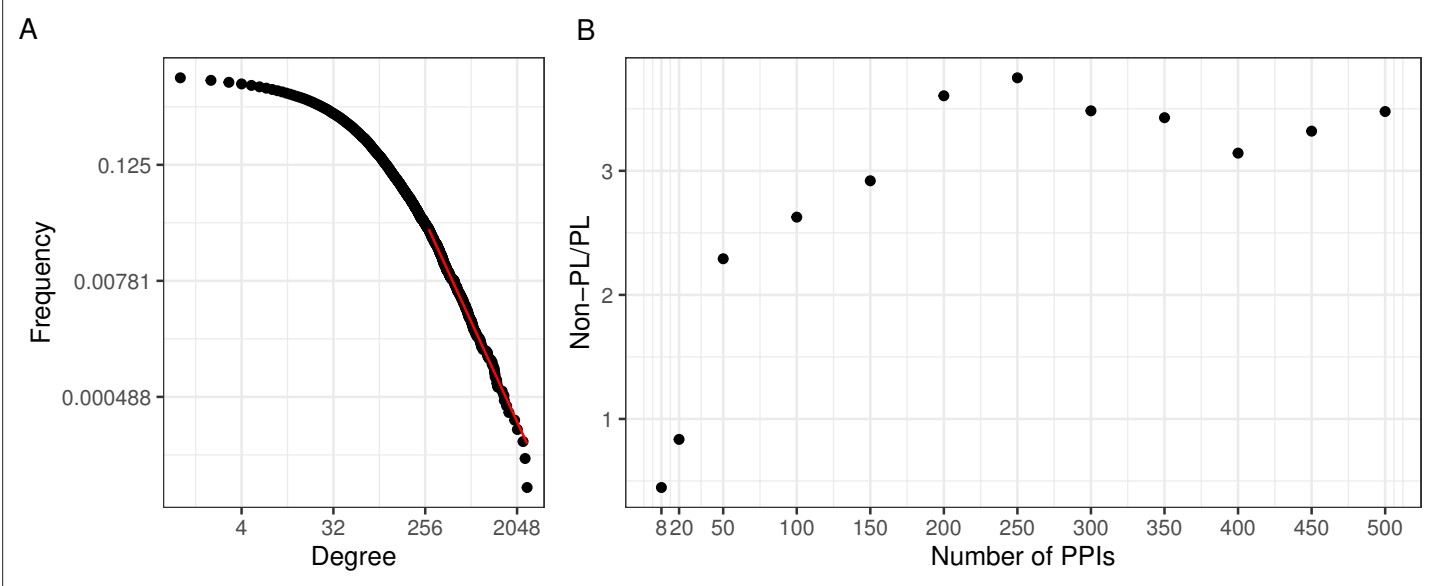

**Figure 2.** A large aggregated PPI network shows PL behavior while individual studies often do not. (**A**) The black dots represent the degree distribution of our aggregated network and the red line corresponds to the fitted power law (PL) distribution with parameters $k_{min} = 278$ and $\alpha = 3.3$ in a log-log scale. (**B**) Plot of the ratio between the number of non-PL and PL studies with more than a certain number of protein-protein interactions (PPIs specified in the $x$-axis).

The online version of this article includes the following figure supplement(s) for figure 2:

**Figure supplement 1.** Histogram of protein-protein interaction (PPI) number per study in the aggregated network.

then quantify to which extent the degree distributions of observed PPI networks allow us to derive conclusions about the topology of the true biological interactome. Overall, our results indicate that technical and study bias can indeed largely explain the fact that observed PPI networks tend to be PL-distributed. This implies that it is problematic to derive hypotheses about the degree distribution and emergence of the true biological interactome from the fact that node degrees in observed PPI networks tend to be PL-distributed.

## Results

### Less than one in three study-specific protein-protein interaction networks are power law distributed

*Mosca et al., 2021* recently showed that aggregated observed PPI networks generally show a node degree distribution following the PL. To confirm this, we aggregated a large human PPI network consisting of 41,862 studies and a total of 471,693 unique interactions among 17,865 proteins. We tested if the resulting degree distribution follows a PL by quantifying the plausibility of a goodness-of-fit test as described before (*Clauset et al., 2009*). We observed that the resulting degree distribution can be approximated by a PL distribution ($p = 0.35$; where $p \geq 0.1$ is by convention (*Clauset et al., 2009*) indicative of a PL distribution; *Figure 2A*).

An interesting question is if the PL property is inherent to single PPI networks or if it possibly arises through the aggregation process. To investigate this, we next tested for the PL property of the constituting single studies. We observed that when considering networks of size 200 or larger, there were approximately 3.5 times as many non-PL-distributed networks as compared to PL-distributed networks (*Figure 2B*). The ratio reduces to 1 when also small networks were considered. However, we reasoned that this is likely an artifact of the relatively poor fit of the degree distribution for small networks: The majority of networks have a small size (*Figure 2—figure supplement 1*) and those small networks that are not filtered out (see Methods), are typically classified as PL-distributed. E. g., 84% of the 739 single-study networks with at most 20 PPIs (a network size which we consider unlikely to lead to reasonable degree distribution fits) are classified as PL-distributed. This suggests that for

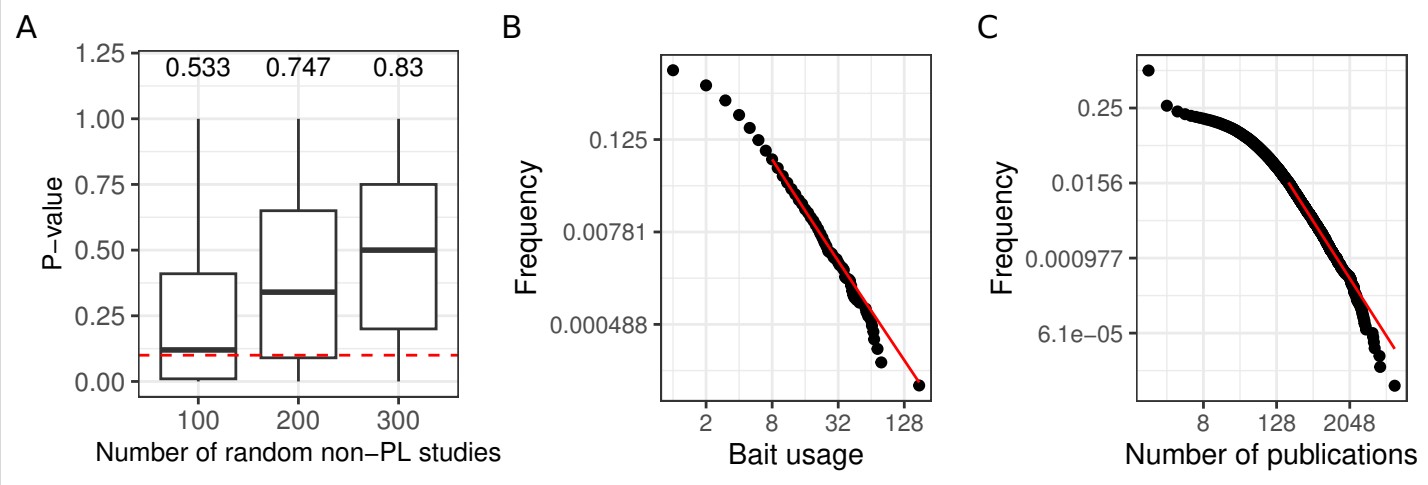

**Figure 3.** Aggregating more PPI networks increases the mean probability of obtaining a PL fit, potentially due to bait usage and study biases. (**A**) Distribution of p-values obtained through the aggregation of 100, 200, and 300 random non-power law (PL) studies. The numbers on the top of each boxplot represent the fraction of PL networks obtained among the 1000 tests. The dotted red line represents the limit of significance (i.e. 0.1); above the line, the PL hypothesis is plausible. (**B**) The black points correspond to the bait usage distribution and the red line corresponds to the fitted PL distribution (in a log-log scale) with parameters $k_{min} = 8$ and $\alpha = 3.13$. (**C**) The number of publications indexed in PubMed associated with different human genes follows a PL distribution ($k_{min} = 201$ and $\alpha = 2.53$).

network sizes where it is possible to reliably fit degree distributions, the non-PL networks largely outnumber PL networks.

### The power law property is associated with research interest

We next systematically tested if the PL property of networks can emerge from aggregating non-PL networks. We, therefore, randomly merged non-PL networks (1000 times 100, 200, and 300 non-PL studies). As shown in *Figure 3A*, we obtained more than 50% of PL aggregated networks after the aggregation of non-PL studies. In particular, the more studies we merged, the greater the fraction of PL studies (from 53 to 83%), demonstrating that the PL property can emerge from the aggregation of non-PL networks.

We observed a good correlation between the number of times a protein has been tested for interaction partners as a bait protein and its degree ($r = 0.57$, $p < 10^{-16}$; Pearson correlation test) in agreement with what has been previously described (*Schaefer et al., 2015*). This raises the question whether the PL property of the merged network could have been inherited from a potential bias in the number of times proteins have been tested for interaction partners. Indeed, we observed that the bait usage distribution follows a PL ($p = 0.34$; *Figure 3B*). To test if the bait usage distribution could impact the observed degree distribution, we randomly subsampled networks for which we have bait information 3000 times (1000 times 50, 100, and 150 non-PL studies). For each resulting aggregated network, we fitted PL distributions to both the bait usage distribution and the degree distribution. We observed a significant association between finding that if one of the distributions follows a PL distribution the other one would tend to do so as well ($p = 0.04$; one-sided Fisher's exact test). To more broadly investigate the association between the PL property and research interest in proteins, we also counted the numbers of publications indexed in PubMed linked to different human genes. Again, the obtained distribution follows a PL ($p = 0.23$; *Figure 3C*).

### Power law distributed research interest can explain the power law property of protein-protein interaction networks

The descriptive findings summarized in the previous paragraphs indicate that the PL property in aggregated PPI networks may reflect biases in the PPI measurement process, instead of capturing the topology of the ground truth interactome. In particular, analyses of the proteins' bait usage counts (*Figure 3B*) as well as their coverage by publications indexed in PubMed (*Figure 3C*) revealed that protein research is itself PL-distributed.

In the following, we mathematically establish that, given PL-distributed bait usage, the degree distribution of an observed PPI network $G_{obs}$ measured via repeated AP-MS testing has to be expected to be PL-distributed, even if the underlying ground truth interactome $G$ has a radically different topology or does not even contain any interaction at all, with observed interactions only being the result of a small false positive error rate. Technically speaking, we establish this fact for the following range of possible interactomes: It is valid for any ground truth interactome $G$ that is a sparse Erdős-Rényi (ER) graph $H_p$ with $n$ nodes, which arises by choosing each of the $\binom{n}{2}$ possible edges with a small edge probability $p \in O(n^{-1})$ uniformly at random (**Erdős and Rényi, 1959**). The degree distribution of these graphs is known to follow a binomial distribution, not a PL. We show that a small false positive rate $FPR \in O(n^{-1})$ and selection bias via a PL-distributed bait usage will result in an expected degree distribution in $G_{obs}$ that follows a PL. More precisely, we show the following *Proposition 1* (see *Methods* for proofs).

*Proposition 1.* Let $G_{obs} = (V, E_{obs})$ be an observed PPI network on $n = |V|$ nodes, which is constructed via aggregated AP-MS testing of the unknown ground truth interactome $G = (V, E)$ as follows: Each protein $u \in V$ is selected $b(u)$ times as bait, and each time $u$ is selected, all of its possible connections $\{uv \mid v \in V\}$ are tested. An edge $uv$ is added to $E_{obs}$ if it is tested positive at least once. The individual AP-MS studies have fixed false-positive and false-negative rates FPR and FNR. Then, if $G$ is an ER graph with edge probability $p$, the expected degrees of $G_{obs}$ are PL-distributed, following the bait usage distribution $b$, if $FPR \in O(n^{-1})$ and $p \in O(n^{-1})$ are small and $n \gg 1$ is large. In particular, this remains true for $p = 0$, where the ground truth is the empty graph.

*Proposition 1* demonstrates that we may observe a PL degree distribution in $G_{obs}$, even if the ground truth $G$ does *not* have such a distribution. *Proposition 1* does of course *not* prove that the ground truth network does *not* follow a PL distribution. However, it demonstrates that stronger arguments than just a PL-distributed observed network $G_{obs}$ are necessary. To exemplify *Proposition 1*, we simulated the simplified aggregated AP-MS testing protocol assumed by *Proposition 1*, using the real-world distribution $b$ obtained from IntAct (**Figure 3B**), an empty ground truth interactome $G = H_0$,

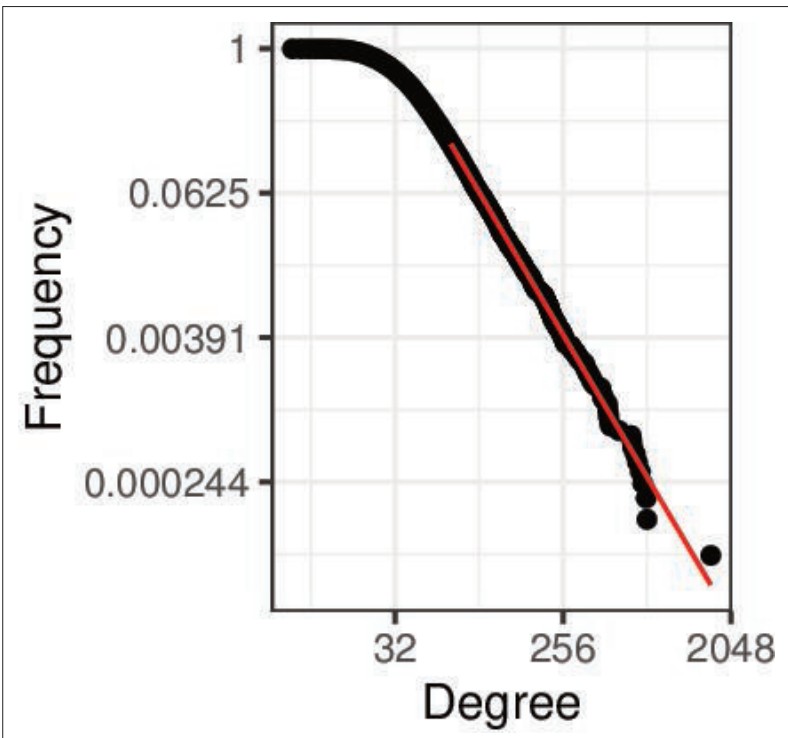

**Figure 4.** Exemplification of *Proposition 1* for an empty ground truth interactome, a small positive error rate, and the real-world bait distribution $b$ obtained from IntAct. The simulated observed degree distribution is power law (PL)-distributed with parameters $k_{min} = 64$ and $\alpha = 3.63$.

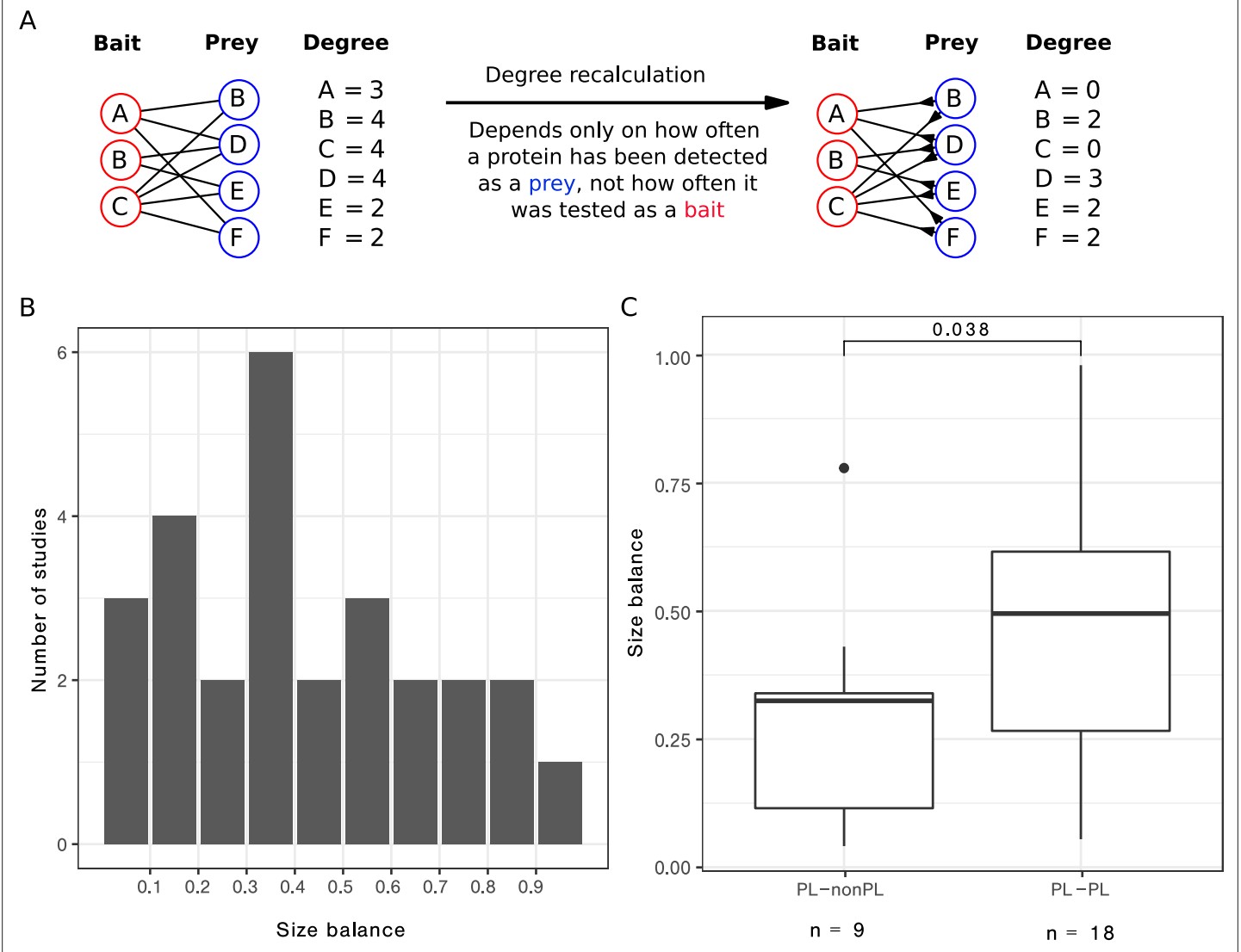

**Figure 5.** After correcting for bait or prey usage, a third of the PL networks lose the PL property. (**A**) Scheme to illustrate how the degree is recalculated when the number of baits is smaller than the number of preys. (**B**) Distribution of the size balance (ratio between the number of baits and preys, see *Equation 2* for details) among the 27 power law (PL) studies. (**C**) Distribution of the size balance in the nine studies that switch from PL to non-PL and the 18 studies whose degree distributions remain PL-distributed after the correction.

and parameters $n = 16777$ (numbers of proteins in IntAct) and FPR = 1/1700 (such that the expected number of edges in $G_{obs}$ matches the number of PPIs retrieved from IntAct). *Figure 4* shows the obtained degree distribution of $G_{obs}$. The observed degrees are PL-distributed ($p = 0.35$), although the underlying ground truth interactome $G$ is empty.

## The power law property often vanishes when correcting for bait usage

We focused on the 27 single-study networks with PL distribution that consisted of more than 200 PPIs (as our initial analysis suggested that the ratio between non-PL and PL studies converges from this value) and for which we had bait and prey information. We observed that the ratio between baits and preys usage varied largely across those studies (*Figure 5B*), resulting in some studies with a symmetric design (i. e. relatively similar number of baits and preys) and some with rather asymmetric design (i. e. big differences in the numbers of baits and preys). We hypothesized that strongly uneven bait vs. prey usage may contribute to the PL property by inflating the degree of a few proteins, effectively favouring them to become hub proteins. To test this hypothesis, we attempted to correct this bias to check whether it is possible to transform PL networks into non-PL networks, solely by correcting for

study bias. To this end, we recomputed the degree distribution by only considering the number of interactions formed by the larger set (either baits or preys, see Methods for details and *Figure 5A* for a graphical visualization).

We observed that in 9 out of the 27 cases, we turned PL degree distributions into non-PL degree distributions by applying this correction. We observed that symmetry scores for networks that changed from PL to non-PL were significantly smaller ($p = 0.038$, one-sided Wilcoxon test; *Figure 5C*), demonstrating that the bait-to-prey ratio has a considerable influence on the PL property.

## Accounting for study bias reveals functionally meaningful hub proteins

The described observations make it critically important to understand how far the degree distribution of proteins is inflated by study bias: To what extent is the degree of proteins with a high degree in the aggregated PPI networks not primarily an indication of proteins with a higher number of interactions, but mainly a result of more frequent testing due to their relevance in disease or other assumed importance in cellular systems? We hence asked if we could reveal the true identity of hub proteins. To this end, we employed three different strategies:

- We computed the degree using only interactions formed by preys in AP-MS studies (with more than 100 PPIs) and identified those with the largest degree (similar to the previous section and visualized in *Figure 5A*; prey hubs).
- We normalized the degree in our initial aggregated network by the number of times the proteins have been used as bait and identified the proteins with the highest normalized degree (normalized hubs).
- We computed the degree distribution within one single study (HuRI *Luck et al., 2020*) that aims to provide a study-bias-free, near-proteome-scale map of the human interactome. We refer to the proteins with the highest degree in this network as Y2H hubs.

We then tested the top 50 hubs for functional (Gene Ontology) and disease gene (Disease Ontology) enrichment. Interestingly, we observed that the prey hubs are most strongly enriched for 'protein folding' and 'chaperone-mediated protein folding' (*Figure 6A*, *Figure 6—source data 1*). The majority of genes in these categories are chaperones whose function is to mediate protein folding. Since the majority of human proteins require assistance in folding by chaperones (*Fink, 1999*), they might indeed be true hub proteins. As chaperones compose 10% of the cellular proteome mass in humans (*Finka and Goloubinoff, 2013*), we were concerned that the enrichment of chaperones among the prey hubs could be an artifact of a detection bias of AP-MS toward highly abundant proteins. To rule out this possibility, we retrieved MS quantifications of human proteins in different tissues (*Jiang et al., 2020*) and performed a Gene Ontology enrichment analysis on sets of the most abundant protein (of different sizes). None had chaperones among the top-enriched terms (*Figure 6—figure supplement 1*), suggesting that the enrichment among prey hubs was not simply an artifact of protein abundance. Similarly, pathway enrichment analysis (*Figure 6—figure supplement 2*) showed protein folding among the top enriched pathways.

The disease gene enrichment analysis confirms the previous observation that uncorrected hubs are associated with many different types of diseases (*Figure 6—source data 1*), in particular with cancer (*Figure 6B*). Prey hubs exhibit an enrichment of diseases related to the nervous system (though much weaker as compared to the enrichment of cancer among the uncorrected hubs). In contrast, normalized and Y2H hubs do not show any significant enrichment in diseases, challenging the idea that disease genes per se have a higher connectivity in PPI networks.

We were surprised to find several nervous system diseases enriched among the prey hubs. Many of the prey hubs related to nervous system diseases were in fact chaperones. To test if the enrichment of nervous system diseases were caused by the chaperones, we retrieved the proteins of the most strongly enriched disease classes (schizophrenia and psychotic disorder) and tested if chaperones were enriched among those proteins (*Figure 6—figure supplement 3*). Indeed, we found a significant enrichment ($p < 0.05$, one-sided Fisher test) in both cases, suggesting that chaperones might cause the observed disease enrichment among true hubs toward the nervous system diseases. This is likely because protein misfolding is a hallmark of many nervous system diseases (*Tittelmeier et al., 2020*; *Nucifora et al., 2019*) and indeed chaperones play a role in the prevention of misfolding.

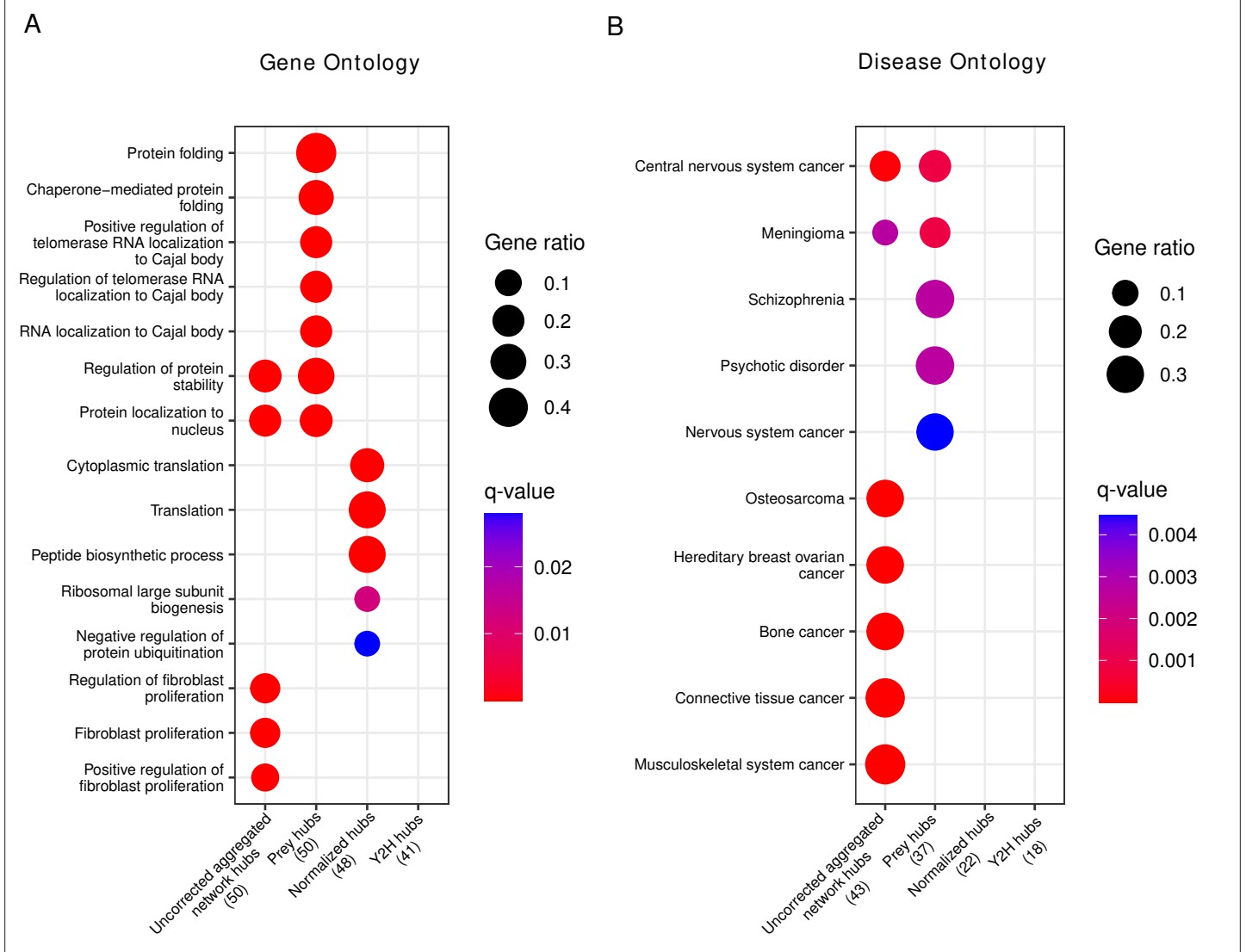

**Figure 6.** Gene set enrichment analysis of hub proteins after bias correction yields biologically plausible terms that differ from uncorrected analysis. (**A**) Gene ontology enrichment analysis of the top 50 corrected (prey hubs, normalized hubs, and Y2H hubs) and non-corrected hubs (uncorrected aggregated network hubs). (**B**) Disease ontology enrichment analysis of the top 50 corrected and non-corrected hubs. The numbers in parentheses represent the number of hubs included in the reference databases, and the 'Gene ratio' represents the fraction of hubs included in the corresponding (gene or disease ontology) term. If a column is empty, it means there are no significant terms.

The online version of this article includes the following source data and figure supplement(s) for figure 6:

**Source data 1.** Detailed results of gene set enrichment analysis.

**Figure supplement 1.** Gene Ontology enrichment analysis results of the top-200, top-500, top-1000, top-2000, and top-3000 most abundant proteins.

**Figure supplement 2.** Reactome enrichment analysis of the top 50 corrected and uncorrected hubs.

**Figure supplement 3.** Overlaps between chaperones and genes related with, respectively, schizophrenia and psychotic disorders.

## Similarity of simulated to observed networks does not depend on the topology of ground truth network

To further assess if the PL property in aggregated PPI networks might be due to biases in the PPI measurement process, we simulated the measurement of observed aggregated PPI networks under preferential interaction testing (see Methods for details). We parameterized our simulator with four hyper-parameters: The test method (AP-MS or Y2H testing), the false positive, and the false negative rates of the test method, and the acceptance threshold $\gamma \in [0, 1]$ (our simulator includes a PPI $(u, v)$ into the simulated aggregated network if it has been detected at least once and the fraction

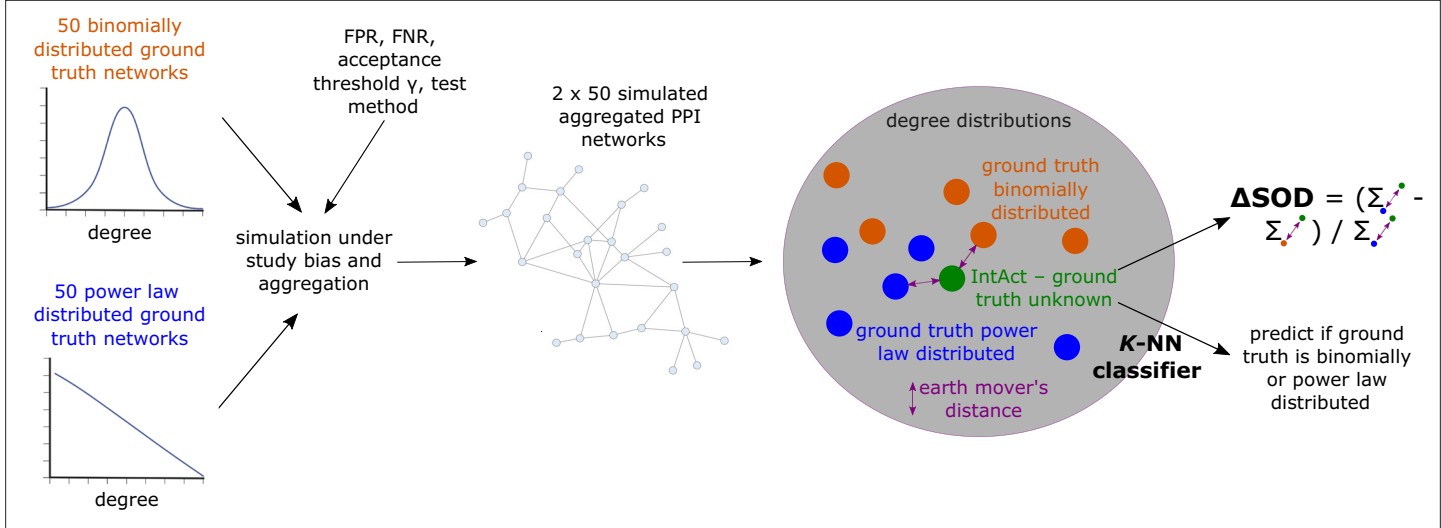

**Figure 7.** Conceptual overview of simulated aggregated protein-protein interaction (PPI) testing under study bias and downstream analyses to assess if the empirical aggregated PPI network $G_{\text{IntAct}}$ obtained from IntAct is more likely to have emerged from a power law (PL)-distributed than from a binomially distributed true biological interactome. The colored dots in the gray area represent degree distributions; dissimilarity between degree distributions is quantified using the earth mover's distance.

of positive simulated experiments that test for $(u, v)$ exceeds $\gamma$). For each hyper-parameter setup, we simulated 50 hypothetical ground truth networks generated with the Barabasi-Albert (BA) model (*Barabasi and Albert, 1999*) and 50 hypothetical ground truth networks generated with the ER model. In BA networks, node degrees are PL-distributed; in ER networks, they follow a binomial distribution. Subsequently, we simulated the detection of the PPIs in the hypothetical ground truth networks via aggregated PPI testing under study bias.

For each hyper-parameter setup, we hence obtained sets $\mathcal{G}_{BA}$, and $\mathcal{G}_{ER}$ which each contains 50 simulated aggregated PPI networks that have emerged from, respectively, PL-distributed ($\mathcal{G}_{BA}$) and binomially distributed ($\mathcal{G}_{ER}$) hypothetical ground truth networks. We then asked if the degree distribution of the empirical aggregated PPI network $G_{\text{IntAct}}$ obtained from IntAct is more similar to the degree distributions of the simulated aggregated networks contained in $\mathcal{G}_{BA}$ than to the degree distributions of the networks contained in $\mathcal{G}_{ER}$. If so, this would indicate that the unknown true biological interactome underlying $G_{\text{IntAct}}$ is more likely to be PL- than binomially distributed. To answer this question, we computed earth mover's distances between the degree distribution of $G_{\text{IntAct}}$ and the degree distributions of the networks contained in $\mathcal{G}_{BA}$ and $\mathcal{G}_{ER}$. Using these distances, we computed a signed relative sum of distance differences $\Delta\text{SOD} \in [-1, 1]$ (*Equation 13*), which is negative if $G_{\text{IntAct}}$'s degree distribution is more similar to the degree distributions of the networks in $\mathcal{G}_{BA}$ than to the degree distributions of the networks in $\mathcal{G}_{ER}$. Moreover, we used the earth mover's distances between $G_{\text{IntAct}}$'s degree distributions and the degree distributions of the networks in $\mathcal{G}_{BA}$ and $\mathcal{G}_{ER}$ to predict via $K$-NN classification if $G_{\text{IntAct}}$ is more likely to have emerged from a PL-distributed or from a binomially distributed true biological interactome (see *Equation 14* and *Equation 15* for details and *Figure 7* for a conceptual visualization).

The results of our simulation studies for AP-MS testing are shown in *Figure 8*. When comparing the empirical network $G_{\text{IntAct}}$ to hypothetical PL-distributed and binomially distributed ground truth networks, we observe that $G_{\text{IntAct}}$'s degree distribution is much more similar to the degree distributions of the PL-distributed networks (*Figure 8A*). This is not surprising, given that $G_{\text{IntAct}}$ is itself PL-distributed (*Figure 8—figure supplement 1A*). However, the picture changes when looking at the sums of distances between $G_{\text{IntAct}}$ and the simulated aggregated networks: Already for very small false positive rates, the gain in similarity between $G_{\text{IntAct}}$ and networks emerging from PL-distributed ground truth networks vanishes. For $\gamma = 0$ (each PPI detected by at least one simulated study is included in the aggregated network), the tipping point lies between FPR = 0.00625 and FPR = 0.0125 (*Figure 8B*); for $\gamma = 0.5$ (a PPI is included in the aggregated network only if it is detected by the majority of the simulated studies that test for it), it lies between FPR = 0.0125 and FPR = 0.025 (*Figure 8C*). By increasing

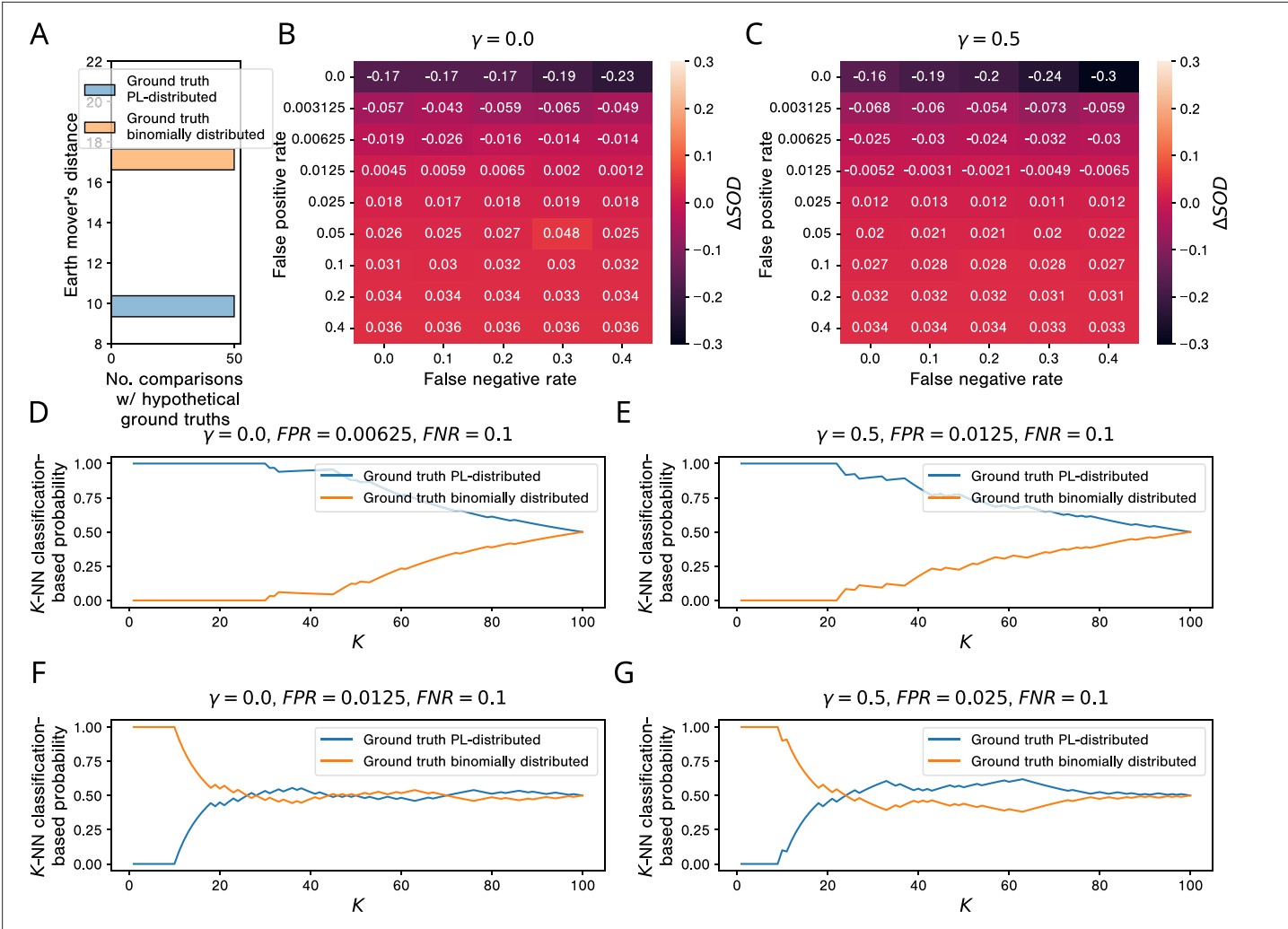

**Figure 8.** Simulations show that, in the presence of study bias and small non-zero false positive rates in affinity purification-mass spectrometry (AP-MS) studies, binomially and PL-distributed ground truth interactomes are equally likely origins of observed aggregated PPI networks. (**A**) Histogram of earth mover's distances between the degree distribution of the observed protein-protein interaction (PPI) network $G_{\text{IntAct}}$ obtained via aggregation of all AP-MS studies annotated in IntAct and the degree distributions of 50 PL-distributed and 50 binomially distributed hypothetical ground truth networks. (**B, C**) Signed relative differences $\Delta\text{SOD} \in [-1, 1]$ between the sum of distances between the degree distribution of $G_{\text{IntAct}}$ and degree distributions of networks simulated from, respectively, power law (PL)-distributed and binomially distributed hypothetical ground truth networks, given different choices of the hyper-parameters FPR, FNR, and $\gamma$. Negative values of $\Delta\text{SOD}$ indicate that $G_{\text{IntAct}}$ is more similar to simulated networks emerging from PL-distributed hypothetical ground truths; positive values are indicative of the opposite scenario. (**D-G**) $K$-NN classification-based probabilities that $G_{\text{IntAct}}$ emerged from a PL-distributed or from a binomially distributed ground truth interactome. (**D, E**) Probabilities just before the tipping points in the false positive rate. (**F, G**) Probabilities just after the tipping points.

The online version of this article includes the following figure supplement(s) for figure 8:

**Figure supplement 1.** Degree distributions of all affinity purification-mass spectrometry (AP-MS) and yeast-2-hybrid (Y2H) studies annotated in IntAct.

**Figure supplement 2.** Results of simulation study for yeast-2-hybrid (Y2H) testing.

$\gamma$ and keeping only consensus PPIs in our simulated networks, we can hence slightly improve the robustness of our simulated PPI network measurement process w. r. t. the false positive rate of the PPI detection method.

$K$-NN classification-based probabilities for $G_{\text{IntAct}}$ having emerged from a PL-distributed or a binomially distributed ground truth interactome for false positive rates just below and just above the tipping points are shown in *Figure 8D* to *Figure 8G*: For false positive rates below the tipping points, a PL-distributed ground truth interactome is clearly the more likely origin of $G_{\text{IntAct}}$, independently of the parameter $K$ used for the $K$-NN classification. For false positive rates above the tipping points and

$K \geq 18$, binomially distributed and PL-distributed interactomes are roughly equally probable origins of $G_{\text{IntAct}}$. With smaller $K$, the estimated probabilities are actually larger for binomially distributed ground truth networks.

If the estimated AP-MS false positive rates of 10 to 40% (*Armean et al., 2013*) are only remotely realistic, they clearly exceed the tipping points between 0.625% and 2.5% uncovered by our simulation study. The results summarized above hence indicate that the observed PL behavior of empirical PPI networks obtained via aggregation of AP-MS studies tells us very little about the topology of the ground truth interactome and is even compatible with binomially distributed node degrees in the ground truth interactome.

The results for simulated Y2H testing (*Figure 8—figure supplement 2*) are very similar to the ones for AP-MS testing. The only difference is that we observe even smaller tipping points. A likely explanation for this is that, unlike the PPI network obtained by aggregating all AP-MS studies annotated in IntAct, the PPI network obtained by aggregating all Y2H studies is itself not PL-distributed (Figure 8—figure supplement 1B). For both simulated AP-MS and simulated Y2H testing, the false negative rate did not have a strong effect on the results (see small row-wise variances in the heatmaps shown in *Figure 8*, *Figure 8—figure supplement 2*).

## Discussion

It is widely believed that the PL behavior of PPI networks arose through evolution, where frequent gene duplication events have led to protein copies that retain the original interaction partners. As one can mathematically prove, such a model eventually leads to a scale-free network (*Chung et al., 2003*). Recently, doubts have emerged that PPI networks are truly scale-free (*Broido and Clauset, 2019*). Furthermore, it was shown that active module discovery methods perform equally well on real and random networks in which the node degree is preserved (*Lazareva et al., 2021*). Such methods, which are typically applied to PPI networks to extract disease modules in the form of subnetworks, thus do not benefit from the interactions of the network but merely learn from the node degree, suggesting that study bias may be driving these analyses.

Here, we offer an alternative explanation, demonstrating that the PL behavior of PPI networks may emerge through a combination of biases. Firstly, we show that typically used experimental designs display an asymmetry between bait and prey proteins, which may contribute to the PL property. Second, we find that the current practice of aggregating study-based PPI networks tends to introduce a PL behavior of the node degree distribution that is not found in the individual studies. Based on the observation that bait usage counts are PL-distributed, we suspect that aggregating studies emphasize study bias, over-representing proteins frequently used as bait. We show that correcting such biases by the three described methods leads to the emergence of alternative hub proteins that drive the network. Thirdly, we mathematically show that, given PL-distributed bait usage, PL-distributed node degrees in observed PPI networks measured via aggregated AP-MS testing can emerge even if the ground truth interactome is an (empty) ER graph. Fourthly, we show through simulation that, already for very small false positive rates, binomially distributed ground truth networks generated with the ER model are equally likely origins for aggregated observed PPI networks as PL-distributed ground truth networks generated with the BA model. This finding is robust across different parameters for the false positive, the false negative, and the study acceptance rate.

It is important to note that the aggregated AP-MS testing model underlying our theoretical results (*Proposition 1*) is simplified in that we assume study bias to act only on the baits. More precisely, we assume that, in the individual studies, a bait is always tested against the entire proteome. In reality, this is not the case for at least three reasons: Firstly, possible interaction partners of a bait are restricted to the proteins expressed in the employed cell line. Second, several inherent properties of proteins correlate with their degrees. E.g., when considering only the subset of AP-MS studies in the here described network, the mass of a protein and its abundance are positively correlated with its degree ($p < 10^{-16}$; Spearman correlation test; *Appendix 1—figure 1*). Third, some studies use targeted proteomics approaches that restrict a priori which peptides can be detected. Moreover, our theoretical results are obtained by modeling the ground truth network as an ER graph, although it is of course very unlikely that this model correctly describes the biological interactome. We chose the ER model for *Proposition 1* because ER graphs are extremely different from PL-distributed networks, and we wanted to show that, even for such radically different ground truths, PL-distributed bait usage

and non-zero false positive rates can lead to PLs in the observed PPI networks. We hypothesize that similar results could be obtained for more realistic non-PL models such as random geometric graphs (*Pržulj et al., 2004*; *Przulj et al., 2010*) but do not provide a formal proof for this here.

Similarly, also the design of our simulation study is based on two major simplifications: First, we only consider ER and BA networks as possible models for the ground truth interactome, although most likely none of the two models fully captures the topology of the unknown true biological inter-actome. Here, this simplification serves as a conceptual framework which allows us to address the question on the origin of the PL behavior of observed PPI networks via $K$-NN classification. Second, our simulator assumes that study bias affects the emergence of aggregated observed PPI networks via a direct feedback loop (high-degree proteins are preferentially sampled for experimental testing). In reality, the feedback loop is much less direct: Study bias in the emergence of aggregated PPI networks is not only mediated via studies reporting PPIs but also and primarily via more indirect pathways such as over-representation of genes encoding highly studied proteins in gene annotation databases (*Haynes et al., 2018*). It is hence likely that real-world repeated PPI testing is slightly less sensitive to the experimental false positive rates than suggested by our simulations. However, in view of the huge margin between the uncovered tipping points (0.0625 to 2.5%) and the estimated false positive rates in AP-MS and Y2H testing (10–40% according to *Armean et al., 2013*), our conclusion remains valid that the topologies of observed PPI networks have little inferential value w. r. t. the unknown ground truth interactome.

Some analyses in the study focus on the two most commonly used PPI detection methods AP-MS and Y2H, both of which rely on the selection of bait proteins to identify interacting partners. Our findings demonstrate that the frequency with which proteins are tested as bait significantly influences the degree distribution of the aggregated human PPI network. An additional complexity arising in AP-MS studies is that more than two interaction partners can be detected. These $n$-ary interactions are commonly transformed into binary interactions using either the spoke model, which reports all interactions with the bait protein (as used by IntAct, for example), or the matrix expansion model, which reports all pairwise interactions. Both expansion models can, in principle, introduce false positives and it would be interesting to consider the effect of the expansion model choice on the PL property in future work.

Combined, Y2H and AP-MS account for more than 70% of all reported interactions in the network used here. Other, less commonly used methods such as protein structure-based approaches also depends on bait protein selection. Conversely, cross-linking mass spectrometry (XL-MS) does not require bait protein selection. XL-MS employs chemical cross-linkers on protein complexes within human cellular lysates, with mass spectrometry (MS) subsequently identifying peptides in close spatial proximity. Given its low assumed error rates and lack of study bias if applied to a proteome-wide scale, XL-MS has the potential to more accurately represent the true topology of the human PPI network. However, XL-MS studies have focused on specific complexes, organelles, or compartments through sample fractionation or purification (*Graziadei and Rappsilber, 2022*), likely introducing through those choices a similar type of bias that influences the observed degree distribution. Hence, a biased choice is applied and will affect the overall degree distribution in a similar manner as exemplified with other experimental methods here.

We point out several strategies that could help to reduce biases of PPI networks. Employing techniques that can detect PPIs with an FDR as low as 1% (*Lenz et al., 2021*) would considerably reduces the technical bias in detecting PPIs. Our study suggests a tipping point in the FPR at which study bias can no longer be tolerated, but this may in reality be higher or lower than what we anticipate here. It is thus not clear how robust techniques need to be to entirely avoid that study bias distorts the topology of observed PPI networks. However, even the use of an error-free technique would not miti-gate pre-existing study bias, which is not just ingrained into existing PPI networks but also indirectly influences the choice of bait and prey proteins used in future studies. An alternative strategy is thus to systematically and objectively study PPIs without prior evidence for the relevance of a protein, a strategy currently followed by the HuRI project (*Luck et al., 2020*). Our results indicate that also the aggregation of non-PL studies tends to lead to networks with PL property, possibly because study bias present in individual studies is magnified in this process. In view of this, an interesting question for the future will be if the aggregation of study-bias-free studies such as HuRI will still favor the emergence of the PL property.

Finally, there are also cost-effective ways to assess and address biases in PPI networks. For instance, we could show that the problem of study bias can be partially mitigated by relying on the information of prey proteins alone. An interesting observation we made was that accounting for this bias revealed a different set of hub proteins enriched for protein folding rather than disease genes. Further work will be needed to establish if true hub proteins exist in the PPI network and what their role is. For instance, it was previously claimed (*Han et al., 2004*) — and controversially discussed (*Agarwal et al., 2010*) — that the correlation of gene expression values between hub nodes with their interaction partners follow a bimodal distribution, leading to the distinction of the party (high correlation) and date (low correlation) hubs. In the future, it would be interesting to study if the ratio of party and date hubs changes when considering prey degree only. We also encourage the field to report negative interactions, since these could be used to define a reasonable study acceptance rate (ratio of positive and negative interactions, $\lambda$ in our simulation) to limit the distorting effect of the FPR, whereas, in the current practice, even unique false positive interactions may be added to the aggregated PPI network.

In conclusion, our analysis supports the alternative hypothesis that the PL behavior observed in aggregated observed PPI networks cannot be treated per se as biologically motivated as the gene duplication model suggests. We face the issue that we currently have no means to reliably disentangle study and experimental bias in the node degree distribution. Our attempts to remove this bias led to differing results depending on the type of normalization we used. In all three cases, disease-associated proteins were demoted. Only the prey hub normalization revealed a significant functional enrichment where proteins such as chaperones that are involved in protein folding have been significantly enriched. While these results seem plausible, we cannot prove that this normalization indeed corrects for all conceivable forms of bias. Our results hence suggest that further work is needed to either perform additional studies that avoid known sources of bias or to develops a robust normalization that removes known biases from existing networks.

## Methods

### Analyzed protein-protein interaction networks

We retrieved human PPIs from IntAct (*Orchard et al., 2014*) (version of 2022-02-03 on https://ftp.ebi.ac.uk/pub/databases/intact/2022-02-03/psimitab/) and HIPPIE (*Alanis-Lobato et al., 2017*) (version 2.2 on http://cbdm-01.zdv.uni-mainz.de/~mschaefer/hippie/download.php). 78% of the interactions in IntAct are annotated with the information of which protein within the pair was used as a bait and which as a prey during the experimental determination of the interaction. To increase the total number of studies, we expanded the IntAct interactions by merging with HIPPIE. After that, we downloaded a list of all 20,401 human proteins from UniProt (*Consortium, 2023*) (only reviewed entries from Swiss-Prot/UniProt, https://www.uniprot.org/uniprotkb?facets=model_organism%3A9606&query=reviewed%3Atrue, version from December 13, 2022). We kept only interactions where both the proteins are in this list resulting in a network consisting of 471,693 interactions and 17,865 proteins were detected by 41,862 studies.

### Testing the power law property of empirical distributions

In order to test if sequences of the proteins' node degrees or bait usages (numbers of times the proteins have been tested as a bait) are PL-distributed, we used the poweRlaw R package *Gillespie, 2015* (version 0.70.6). The package implements methods proposed by *Clauset et al., 2009*. It estimates the best-fitting PL distribution to the data of the form

$$p(k) \propto k^{-\alpha}, \tag{1}$$

where $\alpha > 1$ is the scaling exponent, $k \geq k_{min}$ is the degree or the bait usage sequence, and $k_{min} \geq 1$ is the cutoff above which the PL distribution is fit to the data. The package estimates the $k_{min}$ via a minimization of the Kolmogorov–Smirnov (KS) statistic and uses a maximum likelihood estimator to choose $\alpha$. Subsequently, it carries out a goodness-of-fit test between the empirical data and the fitted PL model. Here, the KS statistic between the fitted model and the empirical distributions is compared to KS statistics between the fitted model and synthetic distributions sampled from the fitted model. Then, a p-value can be computed as the fraction of distances between the fitted model and the synthetic distributions that exceed the distance between the fitted model and the empirical

distribution. Following the convention introduced by *Clauset et al., 2009*, we consider the PL distribution a plausible model for the empirical data if the p-value of the goodness-of-fit test exceeds $0.1$. In the poweRlaw R package, the p-value can be computed with the bootstrap_p function, which we ran with 100 (default parameter) bootstrap simulations.

We tested the PL property for each single study included in our aggregated network. We discarded studies where the used method failed to estimate the $k_{min}$ and hence could not test the PL hypothesis. Moreover, we filtered out studies for which more than 10% of the 100 bootstrapping simulations failed to produce meaningful results (as pointed out in the documentation of the poweRlaw package, this can occasionally happen if all values in the synthetically sampled distribution are below $k_{min}$). We applied those exclusion criteria to any analysis that required a PL computation. After these two filtering steps, the remaining studies are 1427 in total, of which 986 are PL-distributed ($p \geq 0.1$, goodness-of-fit test).

We retrieved the PubMed IDs in which each gene has been studied from PubMed (*Sayers et al., 2022*) (gene2pubmed file, downloaded on April 19, 2023 from https://ftp.ncbi.nih.gov/gene/DATA/) and we selected studies carried out only in human genes (114,548 in total). We calculated how many publications are associated with each gene and after we tested the resulting distribution for its PL property.

## Aggregation of study-specific protein-protein interaction networks

In order to investigate if the PL property arises through the aggregation process, we randomly aggregated 100, 200, and 300 non-PL studies (of 441 in total) 1000 times and we tested the PL hypothesis of the degree distribution after the aggregation. We used a similar randomization strategy to test if there is an association between the degree and the bait usage distribution: We considered only non-PL studies with bait annotations (184 in total) and we randomly merged 50, 100, and 150 studies 1000 times. For each aggregated network, we tested the PL property of the degree and bait usage distribution. We used the one-sided Fisher's exact test to analyse any significant association between the two distributions.

## Computing the degree distributions based on baits or prey only

To assess if the asymmetry in experimental design (i.e. number of baits and preys) affects the PL property, we focused on the 27 single-study networks with PL distribution, having more than 200 interactions (we removed one study with less than 10 bait-prey-annotated interactions) and for which we had bait and prey information. For each of them, we recalculated the degree distribution as follows: If, in the study under consideration, the number of baits is smaller than the number of preys, we only counted those interactions $(u, v)$ for the degree of $u$ where $u$ was tested as a prey. Like this, the degree of a protein only depends on interactions where it was tested as a prey not where it was tested as a bait. If a protein has been tested only as prey, its degree does not change. For studies with less prey than baits, we proceeded conversely and only counted $(u, v)$ for the degree of $u$ if $u$ has been tested as a bait. In other words, we recomputed the degrees as the prey-degree for studies with fewer baits than preys and as the bait-degree for studies with fewer preys than baits.

After the degree recalculation, we computed the size balance between the number of baits and preys, which is defined as follows:

$$\text{Size balance} = \begin{cases} n^{\text{bait}}/n^{\text{prey}} & \text{if } n^{\text{bait}} \leq n^{\text{prey}} \\ n^{\text{prey}}/n^{\text{bait}} & \text{if } n^{\text{bait}} > n^{\text{prey}} \end{cases} \qquad (2)$$

In order to test if the asymmetric design has an effect on PL property, we compared the size balances of studies that switch from PL to non-PL with the size balances of studies for which also the recomputed degree distributions are PL-distributed, using the one-sided Wilcoxon test.

## Functional and disease properties of proteins

We performed functional and disease enrichment analyses of the top 50 hubs detected by the three strategies proposed to reveal the true hub proteins (prey hubs, normalized hubs, and Y2H hubs) and the top 50 hubs of our aggregated network. We used the *enrichGO* function of the clusterProfiler R package (*Wu et al., 2021*) (version 4.4.4) and the *enrichDO* function of the DOSE R package (*Yu*

*et al., 2015*) (version 3.22.1) to perform the Gene and Disease Ontology analyses, respectively. We also performed pathway enrichment analyses (Reactome-based) using the *enrichPathway* function of the ReactomePA R package (*Yu and He, 2016*) (version 1.40). We used the FDR method to correct p-values and we took into account only terms with a *q*-value < 0.05. For each enrichment analysis, we used the entire lists of genes from which we retrieved our hypothetical true hubs and all the genes in our aggregated network as background genes.

To investigate the biological functions of the most abundant human proteins, we retrieved protein abundance data from GTEx (*Jiang et al., 2020*) (https://gtexportal.org/home/downloads/egtex/proteomics), consisting of 201 samples from 32 normal human tissues. We removed proteins with more than 50% of NA values across all the samples (resulting in 8104 proteins), and we calculated the median abundance for each protein. We ordered the proteins according to the median (descending order) to perform the Gene Ontology enrichment analysis of the most abundant proteins (of different set sizes). We used the FDR method to correct p-values and we took into account only terms with a *q*-value <0.05. To test if there is a significant enrichment of chaperones among nervous system disease genes (in particular for schizophrenia and psychotic disorder), we retrieved the chaperone classification from UniProt and nervous system disease-related genes from Disease Ontology (*Schriml et al., 2019*) database using the DOSE R package (*Yu et al., 2015*).

To study the link between degree and protein properties, we used GTEx expression data and protein mass information from UniProt (by querying the web service provided at https://www.uniprot.org/id-mapping with the UniProt IDs of all proteins within our aggregated network).

## Proof of proposition 1

We start with useful facts on edge probabilities. We begin by noting the lower and upper bounds on edge probabilities, based on the probability $p_{(u,v)}$ of an edge $uv$ occurring with $v$ as bait, and the probability $p_{(v,u)}$ of $uv$ occurring with $v$ as prey.

**Lemma 1.** For any ground truth graph $G$ and an observed PPI network $G_{\text{obs}} = (V, E_{\text{obs}})$, the probability $p_{uv}$ that an edge $uv$ occurs in $G_{\text{obs}}$ satisfies the following chain of inequalities.

$$p_{(u,v)} \leq p_{uv} \leq p_{(u,v)} + p_{(v,u)}. \tag{3}$$

*Proof.* We have $p_{uv} = p_{(u,v)} + p_{(v,u)} - Pr[(u,v) \in E_{\text{obs}} \wedge (v,u) \in E_{\text{obs}}]$. This immediately yields the second inequality $p_{uv} \leq p_{(u,v)} + p_{(v,u)}$ because the last term is upper bounded by 0. Because of $p_{(v,u)} \geq Pr[(u,v) \in E_{\text{obs}} \wedge (v,u) \in E_{\text{obs}}]$, we have $p_{(u,v)} \leq p_{uv}$. This proves the first inequality. □

Note that these preceding statements apply independent of specific assumptions on the involved graphs. In the following, we will focus on more specific settings to demonstrate how non-PL distributions (such as in ER graphs) can give rise to observed PL distributions.

**Lemma 2.** If $G = (V, E)$ is an ER graph $H_p$, then the probability $p_{uv}$ that the edge $uv$ occurs in the observed PPI network $G_{\text{obs}}$ subject to the bait distribution $b$, false negative rate FNR, and false positive rate FPR is

$$p_{uv} = Pr[uv \in E_{\text{obs}}] = p \cdot \left(1 - \text{FNR}^{b(u)+b(v)}\right) + (1-p) \cdot \left(1 - (1-\text{FPR})^{b(u)+b(v)}\right). \tag{4}$$

*Proof.* Because $G$ is an ER graph with edge probability $p$, we have $Pr[uv \in E] = p$ and $Pr[(u,v) \notin E] = 1 - p$. We distinguish two cases:

- Case (i): $uv \in E$, which occurs with a probability of $p$. The edge $uv$ is tested $b(u) + b(v)$ times; $b(u)$ times with $u$ as bait and $b(v)$ times with $v$ as bait. The probability that $uv$ is not tested positive in any of these tests is $\text{FNR}^{b(u)+b(v)}$. Thus, we have $Pr[uv \in E_{\text{obs}} \mid uv \in E] = 1 - Pr[uv \notin E_{\text{obs}} \mid uv \in E] = 1 - \text{FNR}^{b(u)+b(v)}$.
- Case (ii): $uv \notin E$, which occurs with a probability of $1 - p$. Then the probability that $uv$ is not tested positive is $(1 - \text{FPR})^{b(u)+b(v)}$ and the probability that $uv$ is tested positive is $Pr[uv \in E_{\text{obs}} \mid uv \notin E] = 1 - Pr[uv \notin E_{\text{obs}} \mid uv \notin E] = 1 - (1 - \text{FPR})^{b(u)+b(v)}$.

Combined, this yields $Pr[(u,v) \in E_{\text{obs}}] = p \cdot \left(1 - \text{FNR}^{b(u)+b(v)}\right) + (1-p) \cdot \left(1 - (1-\text{FPR})^{b(u)+b(v)}\right)$, as claimed. □

As a special case, this implies the following statement about edges occurring purely because of the bait degree of one of its vertices:

**Lemma 3.** If $G = (V, E)$ is an ER graph $H_p$, then the probability $p_{(u,v)}$ that some edge $uv$ in the observed PPI network $G_{\text{obs}} = (V, E_{\text{obs}})$ occurs as the consequence of testing with $u$ as prey and $v$ as bait is

$$p_{(u,v)} = Pr[(u,v) \in E_{\text{obs}}] = p \cdot \left(1 - \text{FNR}^{b(v)}\right) + (1-p) \cdot \left(1 - (1 - \text{FPR})^{b(v)}\right) \tag{5}$$

Now we consider the case that $G$ is an ER graph with small $p$ and small FPR. For FPR sufficiently small, i.e., $a \cdot \text{FPR} \in o(1)$, the classic first-order binomial approximation of the respective edge probabilities work out as follows:

$$(1 - \text{FPR})^a \approx 1 - a \cdot \text{FPR} \tag{6}$$

With this simplification, we can express the expected degree of nodes as follows:

**Lemma 4.** Assume that $G$ is an ER graph on $n$ nodes with edge probability $p$ and that FPR is small. Then, for each node $v \in V$, the expected degree satisfies

$$(1-p)\text{FPR} \cdot \left((n-1)b(v)\right) \leq \mathbb{E}[\deg(v)] \leq (1-p)\left(\text{FPR} \cdot (n-1)b(v)\right) + A, \tag{7}$$

where $B = \sum\limits_{u \in V} b(u)$ and $A = \left(pn + (1-p)\text{FPR} \cdot B\right)$.

*Proof.* Exploiting linearity of expectation, *Lemma 1*, *Lemma 2*, *Lemma 3*, and **Equation 6**, the expected degree of a node $v$ satisfies

$$
\begin{aligned}
&& (1-p)\left(\text{FPR} \cdot (n-1)b(v)\right) \\
&=& (1-p) \cdot \sum_{u \in V\setminus\{v\}} \left(\text{FPR} \cdot b(v)\right) \\
&\overset{\text{(Equation 6)}}{\approx}& (1-p) \cdot \sum_{u \in V\setminus\{v\}} \left(1 - (1 - \text{FPR})^{b(v)}\right) \\
&\overset{\text{(Lemma 3)}}{\leq}& \sum_{u \in V\setminus\{v\}} P_{(u,v)} \\
&\overset{\text{(Lemma 1)}}{\leq}& \mathbb{E}[\deg(v)] \\
&=& \sum_{u \in V\setminus\{v\}} p_{uv} \\
&\overset{\text{(Lemma 2)}}{=}& \sum_{u \in V\setminus\{v\}} \left(p \cdot \left(1 - \text{FNR}^{b(u)+b(v)}\right) + (1-p) \cdot \left(1 - (1 - \text{FPR})^{b(u)+b(v)}\right)\right) \\
&\leq& \sum_{u \in V\setminus\{v\}} p + \sum_{u \in V\setminus\{v\}} (1-p) \cdot \left(1 - (1 - \text{FPR})^{b(u)+b(v)}\right) \\
&\leq& pn + (1-p) \cdot \sum_{u \in V\setminus\{v\}} \left(1 - (1 - \text{FPR})^{b(u)+b(v)}\right) \\
&\overset{\text{(Equation 6)}}{\approx}& pn + (1-p) \cdot \sum_{u \in V\setminus\{v\}} (b(u) + b(v))\text{FPR} \\
&=& pn + (1-p)\text{FPR} \cdot \left(B + (n-2)b(v)\right) \\
&\leq& \left(pn + (1-p)\text{FPR} \cdot B\right) + (1-p)\left(\text{FPR} \cdot (n-1)b(v)\right), \\
&=& A + (1-p)\left(\text{FPR} \cdot (n-1)b(v)\right),
\end{aligned}
$$

as claimed. □

Under suitable choice of parameters — e.g., $\text{FPR} \in O(n^{-1})$ and $p \in O(n^{-1})$ — this implies that the expected degree of a node $v$ in the observed network $G_{\text{obs}}$ corresponds to the bait usage $b(v)$ with some uniform additive correction corresponding to the average prey degree. For a sufficiently unbalanced PL distribution $b$, this implies that the overall distribution remains PL distributed, as the average bait usage $n^{-1}B$ is dominated by larger bait usages. For instance, the following choices suffice for this kind of behavior:

**Lemma 5.** Let $b(v)$ be PL-distributed with $\alpha = 3.13$ (just as the real-world bait usage distribution obtained from IntAct), $k_{min} = 1$, and $Pr[b(v) = 0] = 0.24$ (corresponding to the fraction of proteins from

IntAct with $b(v) = 0$). Moreover, let $n \gg 1$, FPR $= n^{-1}$, and $G = H_0 = (V, \emptyset)$ be a large ER graph with $p = 0$. Then $\deg(v)$ in $G_{\mathrm{obs}}$ is also PL-distributed with $\alpha = 3.13$.

*Proof.* With $\zeta(x) := \sum_{k=1}^{\infty} k^{-x}$ being Riemann's Zeta function, we have $\sum_{k=1}^{\infty} k^{-3.13} = \zeta(3.13) \approx 1.1782$. Thus,

$$Pr[b(v) = k] = \begin{cases} 0.24 & k = 0 \\ 0.76 \cdot \zeta(3.13)^{-1} \cdot k^{-3.13} & k \geq 1 \end{cases}$$

is a probability distribution following a power law with $\alpha = 3.13$. The average bait degree $n^{-1}B$ of a node $v$ works out to

$$\mathbb{E}[b(v)] = 0.76 \cdot \zeta(3.13)^{-1} \sum_{k=1}^{\infty} k \cdot k^{-3.13} = 0.76 \cdot \zeta(3.13)^{-1} \sum_{k=1}^{\infty} k^{-2.13} = 0.76 \cdot \frac{\zeta(2.13)}{\zeta(3.13)} \approx 0.99.$$

By *Lemma 4*, the expected degree of a node $v$ in $G_{\mathrm{obs}}$ satisfies

$$\frac{n-1}{n} \cdot b(v) \leq \mathbb{E}[\deg(v)] \leq \frac{n-1}{n} \cdot b(v) + A \approx \frac{n-1}{n} \cdot b(v) + 0.99.$$

Because of this uniformity, the actual distribution is tightly distributed around this expected value, so the expected degree distribution follows the distribution of $b$, with a small additive constant that becomes insignificant for larger $b(v) = k$. □

Similar behavior can be demonstrated for small positive $p$. It is straightforward to see that, in this case, the additive term $A$ gets increased by not more than $pn$, e. g., it becomes 1.99 for $p = n^{-1}$. In summary, this yields the claims from *Proposition 1* both for $p = 0$ and for small $p$, summarized as follows:

Proposition 2. For an ER graph $G = (V, E)$, large $n \gg 1$, small FPR $\in O(n^{-1})$ and $p \in O(n^{-1})$, and PL-distributed bait usage $b(v)$ for the nodes, the expected degrees of the observed graph $G_{obs}$ are PL-distributed, following the degree distribution of $b$. In particular, this remains true for $p = 0$, where the ground truth is the empty graph.

## Design of simulation study

We simulated observed aggregated PPI networks $G' = (V, E')$ under study bias and different false negative rates FNR false positive rates FPR and from hypothetical ground truth networks $G = (V, E)$. The hypothetical ground truth networks were generated using the BA model, parameterized with the number of nodes $n$ and the number $m_{\mathrm{BA}}$ of edges added per iteration, and the ER model, parameterized with the number of nodes $n$ and the number of edges $m_{\mathrm{ER}}$. The degree distributions of BA graphs are known to follow the power law, while node degrees in ER graphs are binomially distributed. Details on choices of $n$, $m_{\mathrm{BA}}$, and $m_{\mathrm{ER}}$ are provided at the end of this subsection.

We start the simulation of $G'$ with a network on the nodes $V$ without any edges. Throughout the simulation, we add edges to the network by iteratively sampling lists of protein pairs $L_i \subset V \times V$ and then simulating an experiment which, for all $(u, v) \in L_i$, tests if the proteins $u$ and $v$ interact. The experiment returns a binary flag result$(uv) \in \{0, 1\}$, where 1 encodes '$u$ and $v$ interact' and 0 encodes '$u$ and $v$ do not interact.' The result probabilities depend on whether $uv$ is an edge in the ground truth network $G$, as well as on the false negative and false positive rates:

$$Pr[\mathrm{result}(uv) = 1] = \begin{cases} 1 - \mathrm{FNR} & \text{if } uv \in E \\ \mathrm{FPR} & \text{if } uv \notin E \end{cases} \tag{8}$$

To simulate $G'$, we maintain symmetric matrices $\mathbf{A} = (a_{u,v}) \in \mathbb{N}^{V \times V}$ and $\mathbf{B} = (b_{u,v}) \in \mathbb{N}^{V \times V}$. The entry $a_{u,v}$ of the matrix $\mathbf{A}$ counts the number of times the proteins $u$ and $v$ have been tested for interaction, while $b_{u,v} = \sum_{i=1}^{a_{u,v}} \mathrm{result}(uv)$ counts the number of times the experiments have returned that $u$ and $v$ interact. Both $a_{u,v}$ and $b_{u,v}$ are initially set to 0 and increase during simulation. Note that we always have $b_{u,v} \leq a_{u,v}$. After each simulated experiment, $\mathbf{A}$ and $\mathbf{B}$ are updated. Subsequently, we update the edge set of the simulated network as

$$E' = \{uv \mid b_{u,v} > 0 \land b_{u,v}/a_{u,v} > \gamma\}, \tag{9}$$

where $\gamma \in [0, 1)$ is the minimum required fraction of experiments with positive result. The simulation stops once we have carried out $N$ simulated experiments (see end of this subsection for details on choice of $N$). Note that the simulated experiments are asymmetric (i. e. $L_i$ is a list of ordered pairs) but both the hypothetical ground truth interactome $G$ and the simulated observed aggregated network $G'$ are undirected.

To sample the list $L_i$ of protein pairs to be tested for interaction in the $i^{\text{th}}$ experiment, three hyper-parameters are required: the number of baits $n_i^{\text{bait}} \in \mathbb{N}$, the number of preys $n_i^{\text{prey}} \in \mathbb{N}$, and the test method $M \in \{\text{Y2H}, \text{AP-MS}\}$ (which does not depend on $i$). $L_i$ is constructed as $L_i = B_i \times P_i$, where $B_i \subseteq V$ and $P_i \subseteq V$ are sampled lists of baits and preys, respectively. To construct $B_i$, $n_i^{\text{bait}}$ proteins are sampled without replacement from $V$. A protein $u \in V$ is included in $B_i$ with probability

$$Pr[u \in B_i] \propto \deg_{i-1}(u) + \delta, \tag{10}$$

where $\deg_{i-1}(u)$ is $u$'s node degree in the version of the simulated network $G'$ after $i - 1$ experiments and $\delta > 0$ is a hyper-parameter encoding a baseline probability (set to $\delta = 0.01$ in our simulation study). $Pr(u \in B_i)$ hence increases with increasing node degree in the simulated observed network. This leads to a positive feedback loop in the selection of bait proteins, which models study bias in our simulation study.

Since the selection of bait proteins is influenced by study bias both in AP-MS and in Y2H experiments, we use *Equation 10* independently of the test method $M$. In contrast to AP-MS studies, also preys are actively selected in Y2H studies and are thus also directly subject to study bias. Consequently, we construct $P_i$ by sampling $n_i^{\text{prey}}$ proteins without replacement from $V$, where $u \in V$ is included in $P_i$ with probability

$$Pr[u \in P_i] \propto \begin{cases} 1 & \text{if } M = \text{AP-MS} \\ \deg_{i-1}(u) + \delta & \text{if } M = \text{Y2H} \end{cases}. \tag{11}$$

We carried out our simulations for $M \in \{\text{AP-MS}, \text{Y2H}\}$, $\text{FNR} \in \{0.0, 0.1, \ldots, 0.4\}$, $FPR \in \{0.0, 0.4 \cdot 2^{-7}, \ldots, 0.4 \cdot 2^{-1}, 0.4\}$, and $\gamma \in \{0.0, 0.5\}$. The upper bound 0.4 for FNR and FPR was chosen based on estimates for false positive and negative rates in AP-MS and Y2H experiments found in the literature *Armean et al., 2013*. The values for $\gamma$ were chosen to mirror a scenario where a PPI is included in the aggregated PPI network as soon as it is reported by at least one study ($\gamma = 0.0$), as well as a scenario where only those PPIs $(u, v)$ are included for which the majority of studies testing $(u, v)$ report an interaction ($\gamma = 0.5$). Overall, we hence carried out simulations for 180 configurations $(M, \text{FNR}, \text{FPR}, \gamma)$ of free hyper-parameters.

The remaining hyper-parameters were chosen based on the sizes of observed PPI networks obtained for IntAct. For $M = \text{AP-MS}$, we set the overall number of simulated experiments $N$ to the number of AP-MS studies annotated in IntAct where, for each PPI, information about the roles (bait or prey) of the interacting proteins is available. For each study $i$, $n_i^{\text{bait}}$ is set to the number of unique baits used in the study. The number of preys $n_i^{\text{prey}}$ is set to the number of proteins for which an interaction with at least one of the $n_i^{\text{bait}}$ baits is reported by study $i$. Here, we hence make the simplifying assumption that this set equals the set of *prima facie* detectable preys, given the technical setup of the study $i$. To set the hyper-parameters of the hypothetical ground truth networks $G = (V, E)$, we aggregated the PPIs from all $N$ IntAct AP-MS studies and then set $n$ and $m_{\text{ER}}$ to the numbers of nodes and edges in the aggregated network $G_{\text{IntAct}}$. To ensure that also the ground truth networks generated with the BA model have approximately the same number of edges as $G_{\text{IntAct}}$, we set

$$m_{\text{BA}} = \text{round}\left(\frac{n}{2} - \sqrt{\frac{n^2}{4} - m_{\text{ER}}}\right) \tag{12}$$

and initialized the generation of the BA graph with the star on $m_{\text{BA}} + 1$ node (default in NetworkX). With this initialization, the number of edges in the final BA graph equals $|E| = m_{\text{BA}} + m_{\text{BA}} \cdot (n - (m_{\text{BA}} + 1))$, which implies $|E| \approx m_{\text{ER}}$ if $m_{\text{BA}}$ is chosen as specified in *Equation 12*. For $M = \text{Y2H}$, the hyper-parameters

$N$, $n_i^{\text{bait}}$, $n$, $m_{\text{ER}}$, and $m_{\text{BA}}$ were chosen analogously, and $n_i^{\text{prey}}$ was set to the number of preys used in study $i$ (which, unlike in AP-MS studies, are actively selected in Y2H studies).

For each configuration $(M, \text{FNR}, \text{FPR}, \gamma)$ of free hyper-parameters, we sought to answer the question whether, given $(M, \text{FNR}, \text{FPR}, \gamma)$, the observed PPI network $G_{\text{IntAct}}$ is more similar to simulated networks that emerged from a PL-distributed or from a binomially distributed ground truth. For this, we simulated 50 networks $G'$ from BA ground truths (which we collect in the set $\mathcal{G}_{\text{BA}}$) and 50 networks $G'$ from ER ground truths (which we collect in the set $\mathcal{G}_{\text{ER}}$), using the simulator described above. Next, for each $G' \in \mathcal{G}_{\text{BA}} \cup \mathcal{G}_{\text{ER}}$, we computed the earth mover's distance $\text{EMD}(G_{\text{IntAct}}, G')$ between the node degree distributions of $G_{\text{IntAct}}$ and $G'$, and then computed the normalized signed difference

$$\Delta\text{SOD} = \frac{\sum_{G' \in \mathcal{G}_{\text{BA}}} \text{EMD}(G_{\text{IntAct}}, G') - \sum_{G' \in \mathcal{G}_{\text{ER}}} \text{EMD}(G_{\text{IntAct}}, G')}{\sum_{G' \in \mathcal{G}_{\text{ER}}} \text{EMD}(G_{\text{IntAct}}, G')} \tag{13}$$

between the sum of distances between the observed PPI network $G_{\text{IntAct}}$ and the simulated networks contained in $\mathcal{G}_{\text{BA}}$ and $\mathcal{G}_{\text{ER}}$, respectively. $\Delta\text{SOD}$ is negative if $G_{\text{IntAct}}$'s degree distribution is closer to the degree distributions of simulated networks which emerged from a PL-distributed ground truth rather than from a binomially distributed ground truth. Positive values of $\Delta\text{SOD}$ are indicative of the opposite scenario.

We also addressed the question of whether the observed PPI network $G_{\text{IntAct}}$ is more likely to have emerged from a PL-distributed or from a binomially distributed biological interactome, using a simple $K$-NN classifier. More specifically, we sorted the simulated networks $G' \in \mathcal{G}_{\text{BA}} \cup \mathcal{G}_{\text{ER}}$ in increasing order w. r. t. $\text{EMD}(G_{\text{IntAct}}, G')$, leading to a sorted list of networks $(G'_j)_{j=1}^{100}$. For varying $K \in \{1, 2, \ldots, 100\}$, we then computed

$$Pr[G_{\text{IntAct}} \text{ emerged from PL-distributed biological interactome}] = \frac{1}{K} \cdot \sum_{j=1}^{K} \left[ G'_j \in \mathcal{G}_{\text{BA}} \right] \tag{14}$$

$$Pr[G_{\text{IntAct}} \text{ emerged from binomially distributed biological interactome}] = \frac{1}{K} \cdot \sum_{j=1}^{K} \left[ G'_j \in \mathcal{G}_{\text{ER}} \right], \tag{15}$$

where $[\cdot]$ is the Iverson bracket (i.e. `[true]` $= 1$ and `[false]` $= 0$).

## Code availability

Source code to reproduce the results of the simulation study is available at https://github.com/bionet-slab/ppi-network-simulation (copy archived at *Blumenthal and Lucchetta, 2024*). Source code to reproduce all other analyses is available at https://github.com/martaluc/powerlaw-ppi-network (copy archived at *Lucchetta, 2023*).

## Acknowledgements

We thank Leonard Fekete for his valuable contribution to the simulation that produced the data shown in *Figure 4*. We thank Richard Koll for the discussions and his contributions to a preliminary version of the simulator.

## Additional information

### Funding

| Funder | Grant reference number | Author |
| --- | --- | --- |
| Bundesministerium für Bildung und Forschung | 031L0309A | David B Blumenthal |
| Klaus Tschira Stiftung | 00.003.2024 | David B Blumenthal<br>Markus List<br>Martin H Schaefer |

| Funder | Grant reference number | Author |
|---|---|---|
| Fondazione AIRC per la ricerca sul cancro ETS | MFAG 21791 | Martin H Schaefer |
| Fondazione AIRC per la ricerca sul cancro ETS | Bridge Grant n. 29162 | Martin H Schaefer |
| Ministero della Salute | Ricerca Corrente | Martin H Schaefer |
| Ministero della Salute | 5x1000 funds | Martin H Schaefer |

The funders had no role in study design, data collection and interpretation, or the decision to submit the work for publication.

### Author contributions

David B Blumenthal, Conceptualization, Software, Formal analysis, Supervision, Funding acquisition, Investigation, Visualization, Methodology, Writing – original draft, Project administration, Writing – review and editing; Marta Lucchetta, Data curation, Software, Investigation, Methodology, Writing – original draft, Writing – review and editing; Linda Kleist, Sándor P Fekete, Formal analysis, Methodology, Writing – review and editing; Markus List, Conceptualization, Supervision, Funding acquisition, Investigation, Visualization, Methodology, Writing – original draft, Project administration, Writing – review and editing; Martin H Schaefer, Conceptualization, Data curation, Supervision, Funding acquisition, Investigation, Visualization, Methodology, Writing – original draft, Project administration, Writing – review and editing

### Author ORCIDs

David B Blumenthal ⬥ https://orcid.org/0000-0001-8651-750X
Markus List ⬥ http://orcid.org/0000-0002-0941-4168
Martin H Schaefer ⬥ https://orcid.org/0000-0001-7503-6364

### Decision letter and Author response

Decision letter https://doi.org/10.7554/eLife.99951.sa1
Author response https://doi.org/10.7554/eLife.99951.sa2

## Additional files

### Supplementary files
MDAR checklist

### Data availability

We analyzed only previously published data for this work. To facilitate reproducibility, we deposited the used datasets at https://zenodo.org/record/8288898.

The following dataset was generated:

| Author(s) | Year | Dataset title | Dataset URL | Database and Identifier |
|---|---|---|---|---|
| Blumenthal DB, Lucchetta M, Kleist L, Fekete SP, List M, Schaefer MH | 2023 | Emergence of power law distributions in protein-protein interaction networks through study bias | https://doi.org/10.5281/zenodo.7695120 | Zenodo, 10.5281/zenodo.7695120 |

The following previously published datasets were used:

| Author(s) | Year | Dataset title | Dataset URL | Database and Identifier |
|---|---|---|---|---|
| Alanis-Lobato G, Andrade-Navarro M, Schaefer MH | 2017 | HIPPIE v2.2 | https://cbdm-01.zdv.uni-mainz.de/~mschaefer/hippie/download.php | Human Integrated Protein-Protein Interaction rEference (HIPPIE), v2.2 |

*Continued*

| Author(s) | Year | Dataset title | Dataset URL | Database and Identifier |
|---|---|---|---|---|
| Orchard S, Ammari M, Aranda B, Breuza L, Briganti L, Broackes-Carter F, Campbell NH, Chavali G, Chen C, del-Toro N, Duesbury M, Dumousseau M, Galeota E, Hinz U, Iannuccelli M, Jagannathan S, Jimenez R, Khadake J, Lagreid A, Licata L, Lovering RC, Meldal B, Melidoni AN, Milagros M, Peluso D, Perfetto L, Porras P, Raghunath A, Ricard-Blum S, Roechert B, Stutz A, Tognolli M, van Roey K, Cesareni G, Hermjakob H | 2014 | intact.txt | https://ftp.ebi.ac.uk/pub/databases/intact/2022-02-03/psimitab/ | IntAct Molecular Interaction Database, 2022-02-03/psimitab/intact.txt |

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

## Appendix 1

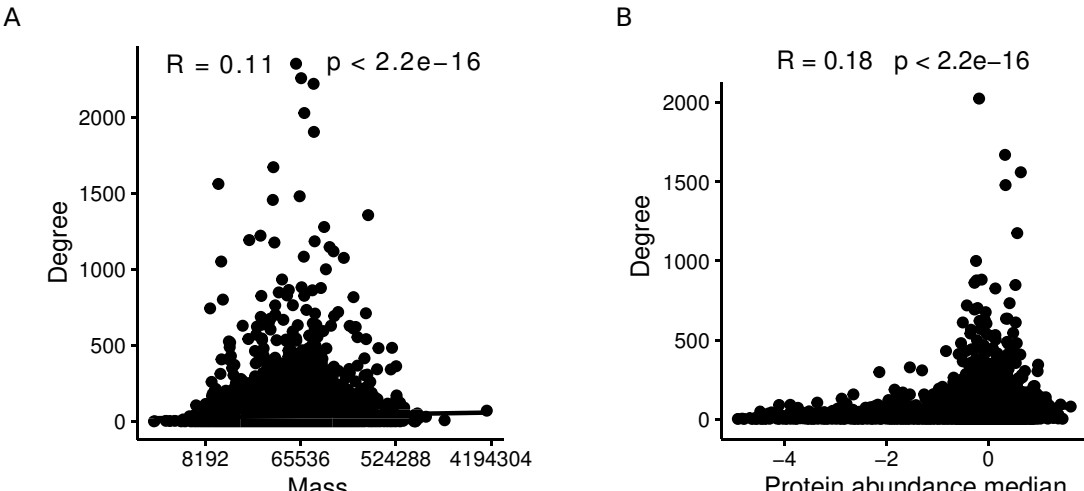

**Appendix 1—figure 1.** Protein mass and abundance are significantly positively correlated with the node degree in aggregated PPI networks. (**A**) Spearman correlation between degree and protein mass within affinity purification-mass spectrometry (AP-MS) studies of the aggregated network. (**B**) Spearman correlation between degree and protein abundance within AP-MS studies of the aggregated network.

