## [Editor Report]

This manuscript makes an important contribution to the understanding of protein-protein interaction (PPI) networks by challenging the widely held assumption that their degree distributions uniformly follow a power law. The authors present convincing evidence that biases in study design, such as data aggregation and selective research focus, may contribute to the appearance of power-law-like distributions. While the power law assumption has already been questioned in network biology, the methodological rigor and correction procedures introduced here help to advance our understanding of PPI network structure.

---

## [Decision Letter]

**Decision letter after peer review:**

Thank you for submitting your article "Emergence of power-law distributions in protein-protein interaction networks through study bias" for consideration by *eLife*. Your article has been reviewed by 3 peer reviewers, and the evaluation has been overseen by a Reviewing Editor and Aleksandra Walczak as the Senior Editor.

We have now received all reviewer feedback on your manuscript, and I am pleased to share that the reviewers are in agreement regarding its merit. All three reviewers find your work to be a valuable contribution to the field and are in favor of publication. They have pointed out a few minor revisions that would improve the overall clarity and precision of the manuscript, particularly in the mathematical formulation. We encourage you to address these points to ensure the best possible presentation of your work.

Please carefully review the detailed comments from the reviewers below. To briefly summarize the main points:

1. Improve the presentation and accessibility for readers, especially in the sections presenting the mathematical results.

2. Clarify the section titled "Similarity of simulated to observed networks does not depend on the topology of ground truth network" to ensure the key ideas are more clearly conveyed.

*Reviewer #2 (Recommendations for the authors):*

My main problem with the paper as it is now is presentational, i.e. the authors make an excessive effort to stress the formal aspects which is only partially successful. The main issue is Proposition 1, which is attempting to be a "mathematical" result but is presented in a vague way ("even if the true graph is ER", 'can be", "FPR […] are small"). This is unhelpful and falls short of a true mathematical result where you'd expect some quantification (e.g. some bounds or asymptotic of probability of misclassification). Such a result would be difficult to prove I think because it also depends on the test used to decide whether a network is actually PL. As it stands, it is a purely existential result which is actually shown via simulation, and that is totally fine: you state the generative mechanism, and you show that under this ER+bias generative mechanism, you get PL in a variety of scenarios (of course you may wish to explore more scenario, like changing the threshold to declare a network PL, but I don't think it is necessary).

*Reviewer #3 (Recommendations for the authors):*

Beyond of what was written in my public report, I would recommend to rewrite the section "Similarity of simulated to observed networks does not depend on the topology of ground truth network". It was difficult to follow and could benefit from a figure showing the concept.

---

## [Author Response]

The reviewers have discussed their reviews with one another, and the Reviewing Editor has drafted this to help you prepare a revised submission.We have now received all reviewer feedback on your manuscript, and I am pleased to share that the reviewers are in agreement regarding its merit. All three reviewers find your work to be a valuable contribution to the field and are in favor of publication. They have pointed out a few minor revisions that would improve the overall clarity and precision of the manuscript, particularly in the mathematical formulation. We encourage you to address these points to ensure the best possible presentation of your work.Please carefully review the detailed comments from the reviewers below. To briefly summarize the main points:1. Improve the presentation and accessibility for readers, especially in the sections presenting the mathematical results.

To improve accessibility, we have provided a more intuitive description in the leadup to Proposition 1. Moreover, we now better explain that some expressions we had used in the formulation of Proposition 1 and which Reviewer 2 had criticized as being too vague for a “true mathematical result” actually do have a well-defined meaning. See reply to first recommendation of Reviewer 2 for details.

2. Clarify the section titled "Similarity of simulated to observed networks does not depend on the topology of ground truth network" to ensure the key ideas are more clearly conveyed.

We have substantially revised this section and have added a figure to visualize the key idea (Figure 7 in revised manuscript). See reply to last recommendation of Reviewer 2 for details.

Reviewer #2 (Recommendations for the authors):My main problem with the paper as it is now is presentational, i.e. the authors make an excessive effort to stress the formal aspects which is only partially successful. The main issue is Proposition 1, which is attempting to be a "mathematical" result but is presented in a vague way ("even if the true graph is ER", 'can be", "FPR […] are small"). This is unhelpful and falls short of a true mathematical result where you'd expect some quantification (e.g. some bounds or asymptotic of probability of misclassification). Such a result would be difficult to prove I think because it also depends on the test used to decide whether a network is actually PL. As it stands, it is a purely existential result which is actually shown via simulation, and that is totally fine: you state the generative mechanism, and you show that under this ER+bias generative mechanism, you get PL in a variety of scenarios (of course you may wish to explore more scenario, like changing the threshold to declare a network PL, but I don't think it is necessary).

Thank you for the request for clarification. We do indeed provide such a mathematical result for the class of sparse Erdős-Rényi (ER) graphs, with stronger implications than just a simulation. We show that, for these graphs, which really have a binomial degree distribution, even just a small positive error rate with PL-distributed selection bias will convert the expected observed degree distribution into one that follows a power law. The expressions “even if G is an ER”, “can be”, and “small” have the following precise meanings:

– “even if G is an ER”: This describes the class of ground truth interactomes to which Proposition 1 applies. We agree that the word “even” is unnecessary here from a technical point of view and have hence removed it from the revised formulation of Proposition 1.

– “can be”: The intended meaning of the statement “the degrees Gobs can be PL-distributed” in the original formulation of Proposition 1 was that the expected degrees of Gobs are PL-distributed (this is the result we establish in our proof). In the revised version of the manuscript, we now explicitly state this in Proposition 1.

– “small”: As stated already in the original version of Proposition 1, this here means that p,FRP∈O(n−1).

For clarification, we have further provided a more intuitive description in the leadup to Proposition 1 that will hopefully clarify its far-reaching implications. The paragraph before Proposition 1 now reads as follows:

“In the following, we mathematically establish that, given PL-distributed bait usage, the degree distribution of an observed PPI network Gobs measured via repeated AP-MS testing has to be expected to be PL-distributed, even if the underlying ground truth interactome G has a radically different topology or does not even contain any interaction at all, with observed interactions only being the result of a small false positive error rate. Technically speaking, we establish this fact for the following range of possible interactomes: It is valid for any ground truth interactome G that is a sparse Erdős-Rényi (ER) graph Hp with n nodes, which arises by choosing each of the n2 possible edges with a small edge probability p∈O(n−1) uniformly at random (Erdős and Rényi, 1959). The degree distribution of these graphs is known to follow a binomial distribution, not a PL. We show that a small false positive rate FPR∈O(n−1) and selection bias via a PL-distributed bait usage will result in an expected degree distribution in Gobs that follows a PL. More precisely, we show the following Proposition 1 (see Methods for proofs).”

Reviewer #3 (Recommendations for the authors):Beyond of what was written in my public report, I would recommend to rewrite the section "Similarity of simulated to observed networks does not depend on the topology of ground truth network". It was difficult to follow and could benefit from a figure showing the concept.

We have re-written large parts of this section and have added a figure to visualize the concept (Figure 7 in revised manuscript). See reply to last comment of Reviewer 2 for more details.